# Myosin VIII associates with microtubule ends and together with actin plays a role in guiding plant cell division

**Shu-Zon Wu, Magdalena Bezanilla\***

Department of Biology, University of Massachusetts, Amherst, Amherst, United States

**Abstract** Plant cells divide using the phragmoplast, a microtubule-based structure that directs vesicles secretion to the nascent cell plate. The phragmoplast forms at the cell center and expands to reach a specified site at the cell periphery, tens or hundreds of microns distant. The mechanism responsible for guiding the phragmoplast remains largely unknown. Here, using both moss and tobacco, we show that myosin VIII associates with the ends of phragmoplast microtubules and together with actin plays a role in guiding phragmoplast expansion to the cortical division site. Our data lead to a model whereby myosin VIII links phragmoplast microtubules to the cortical division site via actin filaments. Myosin VIII's motor activity along actin provides a molecular mechanism for steering phragmoplast expansion.

## Introduction

To divide, plant cells must build a new cell wall. This is accomplished by a dynamic and complex structure known as the phragmoplast, comprising the cytoskeletal polymers, microtubules and actin filaments. The phragmoplast assembles from the anaphase spindle and directs the traffic of vesicles carrying cell wall components to and from the nascent wall, called the cell plate (*Seguí-Simarro et al., 2004*; *Austin et al., 2005*). The phragmoplast forms at the center of the cell and expands outward to reach the parental cell wall, tens or even hundreds of microns distant. When the phragmoplast reaches the parental cell wall, the cell plate and parental membranes fuse, completing cytokinesis.

Active guidance is required during phragmoplast expansion. However, the molecular basis for this steering has been elusive. It is clear that microtubules are essential for cell division, since in their absence the cell plate does not form. And while it has been known for decades that the phragmoplast also contains actin filaments (*Clayton and Lloyd, 1985*; *Kakimoto and Shibaoka, 1987*), it remains unclear how actin contributes to phragmoplast function. For one, plant cells still divide in the absence of actin (*Baluska et al., 2001*; *Nishimura et al., 2003*). Additionally division still occurs in the presence of mutations in genes encoding actin or various actin-associated proteins (*Jürgens, 2005*). However, based on drug treatments and localization studies, actin has been proposed to stabilize the phragmoplast and link the phragmoplast to the cell cortex (*Lloyd and Traas, 1988*; *Molchan et al., 2002*). But beyond implicating actin in a steering mechanism somehow, these studies have provided little if any mechanistic details.

Specifically dissecting the role of phragmoplast actin is further complicated because actin is also present at the preprophase band (*Pickett-Heaps and Northcote, 1966*; *Kakimoto and Shibaoka, 1987*; *Palevitz, 1987*), a microtubule-based structure that is present on the cell cortex just prior to nuclear envelope breakdown. As mitosis proceeds, the microtubules in the band disassemble while actin filaments become depleted from the band itself but enriched on either side of it (*Cleary, 1995*). Although the cytoskeletal polymers are lost from the band, a number of proteins remain at the site of the band throughout mitosis and cytokinesis, thereby marking the site where the new cell plate will

**\*For correspondence:**
bezanilla@bio.umass.edu

**Competing interests:** The authors declare that no competing interests exist.

**Reviewing editor**: Dominique C Bergmann, Stanford University, United States

**eLife digest** Plant cells are surrounded by a membrane, which controls what enters and leaves the cell, and a cell wall, which provides rigidity. When a plant cell is ready to divide, it needs to produce two new cell membranes, with a new cell wall sandwiched between them, to split the cell contents into two daughter cells.

During the division process the cell builds a scaffold called the phragmoplast that guides the delivery of the materials that are needed to make the new cell wall and membranes. The phragmoplast—which is made of rod-like proteins called microtubules and actin filaments—starts at the centre of the cell and expands towards a pre-determined site on the existing cell wall. The question is: how does the phragmoplast target this site, which can be tens or hundreds of microns away?

Wu and Bezanilla have now found that a protein called myosin VIII has a central role in guiding the growing phragmoplast to the cell wall. Myosin VIII is a motor protein that moves along actin filaments. Wu and Bezanilla propose that myosin VIII can guide the expansion of the phragmoplast by pulling microtubules along the actin filaments.

The experiments were carried out on two distantly-related plant species, tobacco and a moss called *Physcomitrella patens*. Similar results were found in both species so it is possible that myosin VIII may play the same role in cell division in all plants.

fuse to the parental cell wall (*Walker et al., 2007*; *Xu et al., 2008*). Interfering with preprophase band development invariably interferes with cell plate positioning (*Rasmussen et al., 2011b*). Because actin is present in both the band and the phragmoplast, discovering actin's function specifically in the latter has been challenging.

However, not all dividing plant cells have a preprophase band. Moss spores germinate into a branched network of filaments, known as protonemata. All dividing cells, both apical and branching, divide without benefit of a preprophase band (*Doonan et al., 1985*). While depolymerization of the actin cytoskeleton halts cell expansion in protonemata, it has little if any effect on cell division. The fact that moss protonemata do not make a preprophase band, but have actin in the phragmoplast provides a unique opportunity to study the role of actin in phragmoplast guidance. Here, we use a combination of genetics and live-cell imaging to probe the role for guiding the phragmoplast of actin and a family of actin-based molecular motors, the class VIII myosins.

## Results

### Cell plate guidance defects in myosin VIII null plants

*Physcomitrella patens* has five identified class VIII myosin genes, named myo8A through E. Taking advantage of facile homologous recombination in this species, *Wu et al. (2011)* constructed a line in which all five genes were disrupted (Δmyo8ABCDE). Protonemata from this line have multiple, unevenly distributed branches. Upon further inspection, we found that cell plate placement at branch sites is often affected (*Figure 1A*). Cell plates are aberrantly positioned with respect to the filament axis (*Figure 1A*, arrows). Since branch patterning and cell division plane specification are linked, we reasoned that non-branching cells in the myosin VIII null plants might also have cell division defects. In young wild-type plants, apical cells position their new cell plates perpendicular to the long axis of the cell: more than 84% of apical cell plates are within 15° of the perpendicular plane. In contrast in myosin VIII null plants, less than 35% of the apical cell plates are within 15° of the perpendicular axis and nearly 40% have cell plates with angles greater than 25°, some as high as 45° (*Figure 1B*).

To investigate how myosin VIII regulates cell plate positioning, we generated a construct encoding Myo8A fused to three tandem copies of monomeric enhanced GFP (hereafter referred to as Myo8A-GFP) and transformed Myo8A-GFP into the myosin VIII null plant. Since myosin VIII's are partially redundant (*Wu et al., 2011*), we reasoned that expression of Myo8A should be sufficient to partially rescue the myosin VIII null phenotype. To test this, we measured cell plate positioning in young plants and found that expression of Myo8A-GFP results in plants with 63% of the apical cell plates within 15° of the perpendicular axis. Importantly, cell plates with angles greater than 35° are never observed in

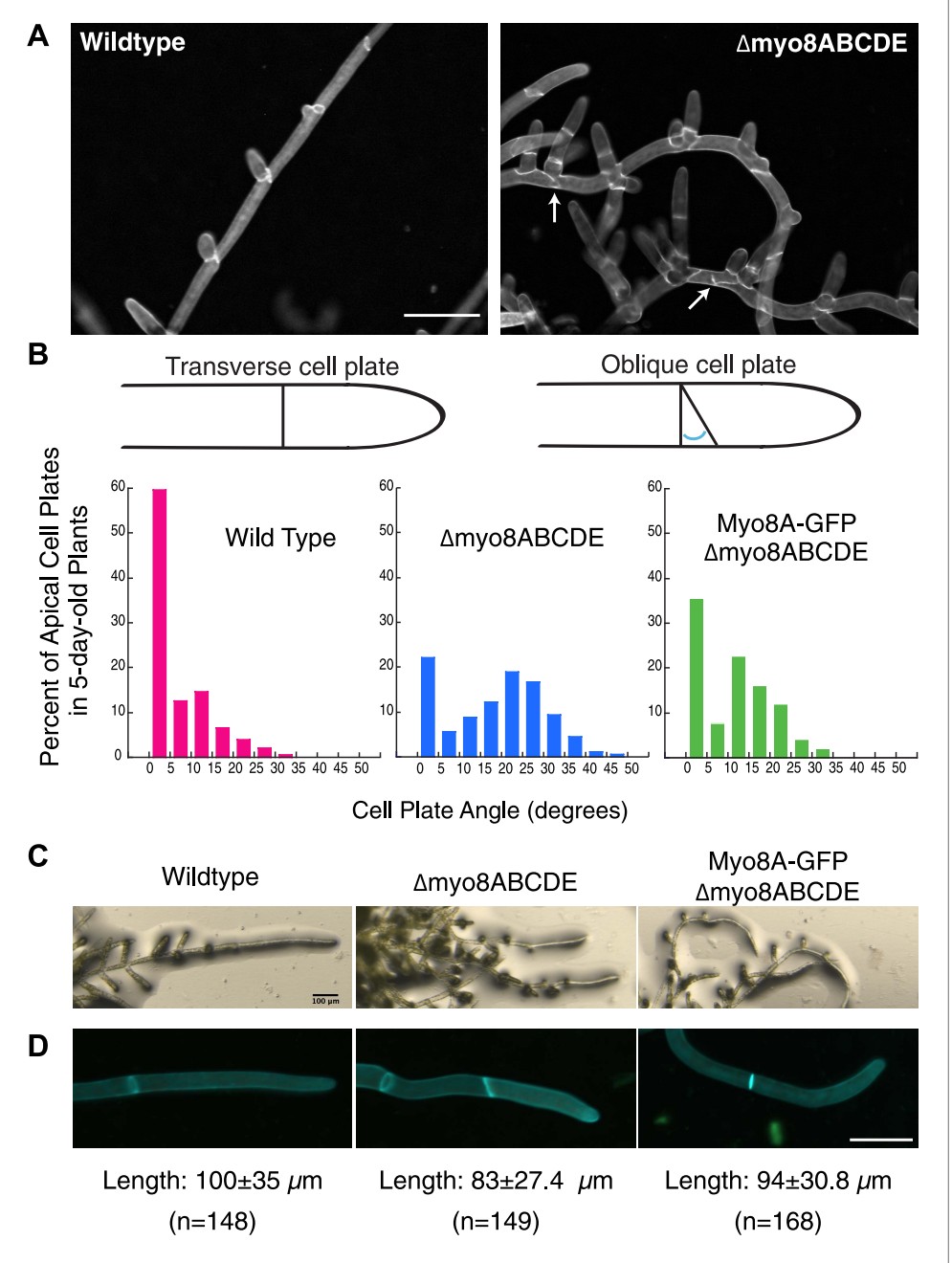

**Figure 1**. Cell plate defects in Δmyo8ABCDE can be restored by expression of Myo8A-GFP. (**A**) 10-day-old wild type and myosin VIII null plants stained with calcofluor. Scale bar, 100 µm. Arrows indicate mis-positioned cell plates. (**B**) Histograms of cell plate angles of apical cells from 5-day-old plants regenerated from protoplasts. Images of apical cells were acquired as in *Figure 1A* and cell plate angles were measured manually using ImageJ. Number of cells analyzed: wild type (n = 151), Δmyo8ABCDE (n = 180), Myo8A-GFP in Δmyo8ABCDE (n = 167). All distributions are significantly different from each other (Wilcoxon-Mann-Whitney Rank Sum Test, p < 0.001). (**C**) 8-day old plants regenerated from protoplasts were imaged with a stereo microscope. Scale bar, 100 µm. (**D**) Measurements of cell length were made on images of the apical cells from calcofluor stained 5 and 6-day old plants regenerated from protoplasts. Average apical cell lengths with standard deviation are indicated below each image. n indicates the number of cells measured. Scale bar, 50 µm.

the Myo8A-GFP expressing plants (*Figure 1B*), indicating that Myo8A-GFP partially restores cell plate positioning in the myosin VIII null plants. Additionally, Myo8A-GFP expression partially rescues a number of other defects in myosin VIII null plants, including protonemal branching defects (*Figure 1C*), apical cell length (*Figure 1D*), and timing of gametophore formation (data not shown). Taken together, our data indicate that Myo8A-GFP is functional.

## Myo8A-GFP moves on actin filaments

Myo8A-GFP localizes diffusely and as small particles throughout the cytoplasm (*Figure 2A*) as well as at the cell cortex (*Figure 2B*). In apical cells Myo8A-GFP particles are enriched at the cell tip (*Figure 2A*). Using variable angle epifluorescence microscopy (VAEM), we simultaneously imaged Myo8A-GFP and actin labeled with lifeact-mCherry. We observed that Myo8A-GFP cortical particles appear to move

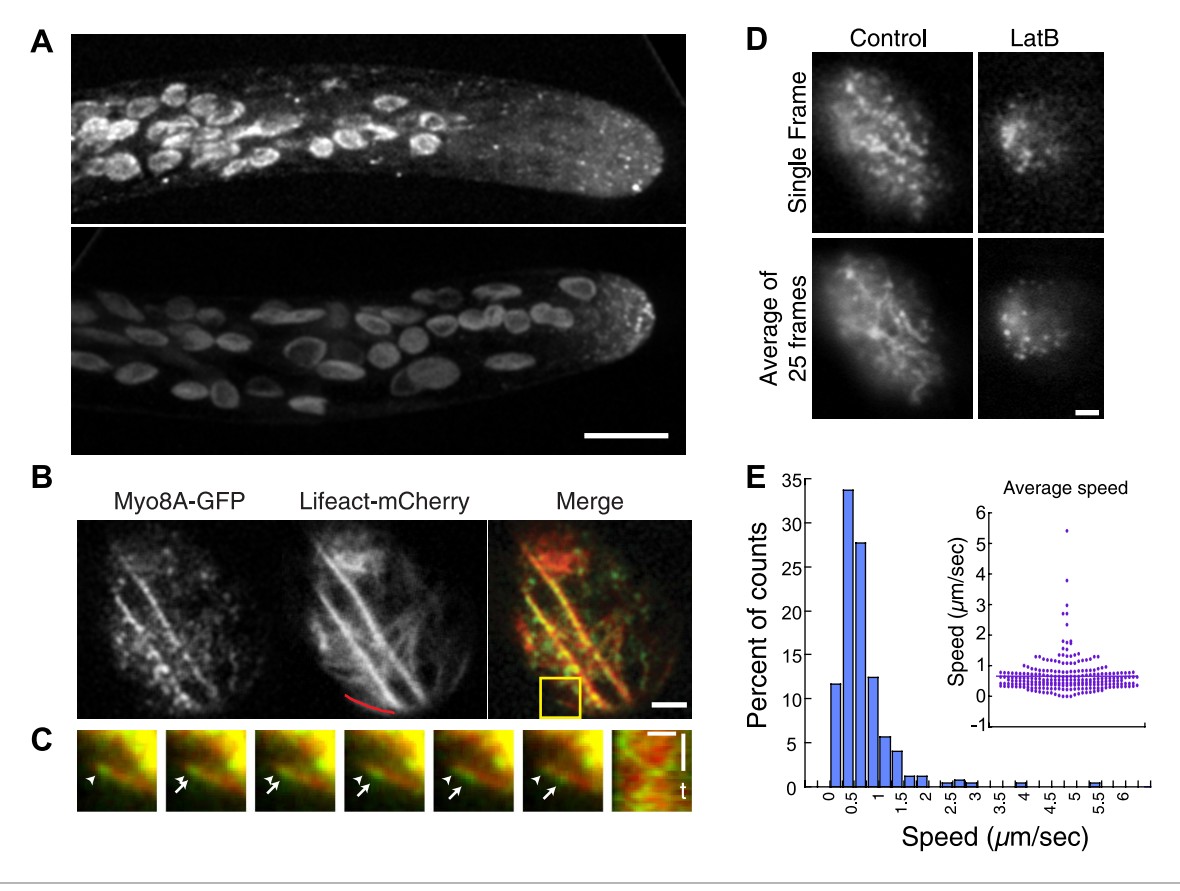

**Figure 2**. Myo8A moves on cortical actin filaments. (**A**) Myo8A-GFP localizes to punctate structures throughout the cytosol as well as on the cell cortex. Images are maximum projections of z-stacks acquired with a spinning disc confocal. The punctate structures accumulate near the apex of the growing cell. Large globular structures are chloroplasts, which autofluorescence under these imaging conditions. Scale bar, 10 μm. (**B**) Images of Myo8A-GFP and Lifeact-mCherry in moss protonemata were simultaneously acquired with VAEM. In the merge Myo8A-GFP is green and Lifeact-mCherry is red. Scale bars, 2 μm. See also *Video 1*. Yellow box indicates the enlarged area shown in (**C**). Red line marks the trace for making the kymograph in (**C**). (**C**) An example of Myo8A-GFP particle moving along actin filaments. Six consecutive frames with 76 ms time interval are shown. Arrowhead indicates the starting position of a Myo8A-GFP particle, and arrows indicate the last position of that Myo8A-GFP. In the last frame, a new Myo8A-GFP particle binds to the same position indicated by the arrowhead. Linear movement of Myo8A-GFP is evident in kymograph. Scale bar, 1 μm. Scale bar in t, 1 s. (**D**) Moss protonemal cells expressing Myo8A-GFP, were treated with or without 25 μm Latrunculin B (LatB) and imaged with VAEM. In control samples, Myo8A-GFP linear trajectories are apparent in a frame average of 25 frames from approximately 2 s of real time, but absent in cells treated with LatB. Scale bars, 2 μm. See also *Video 2*. (**E**) Distribution of Myo8A-GFP velocities on actin filaments. Inset is a dot plot of the measured Myo8A-GFP velocities. Average velocity is 0.65 ± 0.57 μm/s; n = 249 events from seven cells.

The following figure supplement is available for figure 2:

**Figure supplement 1**. Myo8A moves on cortical actin filaments.

along actin filaments, consistent with actin-based motility (*Figure 2B*; *Video 1*). Movement was observed in both directions along an actin cable (*Figure 2C*). We measured the velocity of the particles and found that Myo8A-GFP moves at 0.65 ± 0.57 µm/s (n = 249 particles from 7 cells, *Figure 2E*, *Figure 2—figure supplement 1*). In the presence of 25 µM latrunculin B (LatB), which depolymerizes the actin cytoskeleton, Myo8A-GFP still localizes to the cell cortex but no longer exhibits directed motility (*Figure 2D*; *Video 2*), as expected for an active, actin-based molecular motor.

## Myosin VIII localizes to the phragmoplast and cortical division site

To investigate the role of myosin VIII in cell division, we first examined the localization of Myo8A-GFP in divisions that form branches since the myosin VIII null plants have a strong phenotype in branching cells. To stage division, we introduced into the Myo8A-GFP line mCherry-PpTUA1 (hereafter referred to as mCherry-tubulin). In branch forming cells, Myo8A-GFP accumulates prominently at the cell cortex at the neck region of the emerging bulge, and this accumulation happens prior to prophase (*Figure 3* arrows; *Figure 3—figure supplement 1*; *Video 3*). The tubulin images confirm that these cells lack a preprophase band. During branch cell mitosis, Myo8A-GFP appears on the spindle and phragmoplast, and as the phragmoplast matures, Myo8A-GFP accumulates at the phragmoplast periphery. In the later stages of cytokinesis, there are two populations of Myo8A-GFP: an inner ring on the phragmoplast (*Figure 3*, arrow heads; *Figure 3—figure supplement 1*) and an outer ring at the cell cortex (*Figure 3*, arrows; *Figure 3— figure supplement 1*). As the phragmoplast expands, the inner Myo8A-GFP ring eventually reaches the outer one. Together with an increased frequency of cell plate positioning defects in myosin VIII null plants, these data suggest that myosin VIII plays a role in ensuring that the phragmoplast expands to the pre-determined cortical division site.

To investigate whether myosin VIII has a similar role in seed plants, we generated a tobacco BY-2 cell line stably transformed with the moss Myo8A-GFP. BY-2 cells are commonly used to study the localization of plant proteins during mitosis and cytokinesis (*Van Damme et al., 2004*; *Rasmussen et al., 2011a*; *Lipka et al., 2014*). Alignment of myosin VIII proteins from tobacco, Arabidopsis and moss, revealed that moss myosin VIII proteins are ~50% identical to tobacco and Arabidopsis myosin VIIIs (*Figure 4—source data 1, 2*). Except for the closely related tobacco and Arabidopsis Myo8B and D, the percent identity between the remaining four tobacco and Arabidopsis myosin VIIIs is 51–58% (*Figure 4—source data 1 and 2*). With such high sequence identity, we reasoned that Myo8A-GFP might serve as a proxy for the localization of seed plant myosin VIIIs.

We found that similar to the interphase localization described above for moss, Myo8A-GFP localizes in tobacco to dynamic cortical particles (*Figure 4A*; *Video 4*). In cells about to enter mitosis, cortical Myo8A-GFP accumulates at the

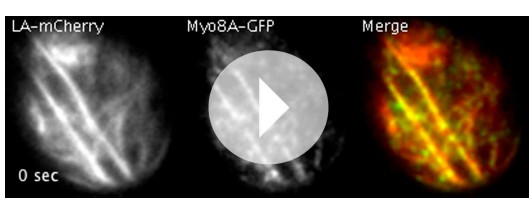

**Video 1**. Myo8A moves on cortical actin filaments (see *Figure 2B*). Myo8A-GFP (green) and Lifeact-mCherry (red) were simultaneously imaged with VAEM (acquired at 13 fps). Video is playing at 15 fps. Scale bar, 2 µm.

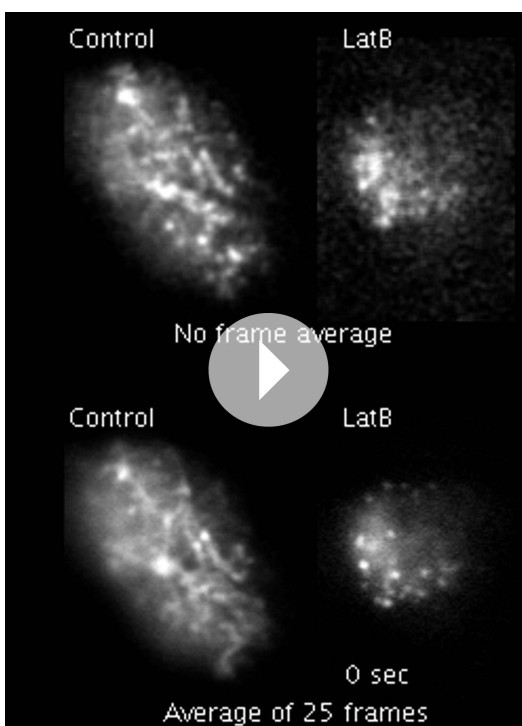

**Video 2**. Myo8A motility at the cell cortex depends on actin (see *Figure 2D*). Moss protonemal cells expressing Myo8A-GFP were imaged with VAEM continuously at 12.233 fps. Video is playing at 25 fps. Scale bar, 2 µm. Bottom panels show the same time series with a 25 frame rolling average.

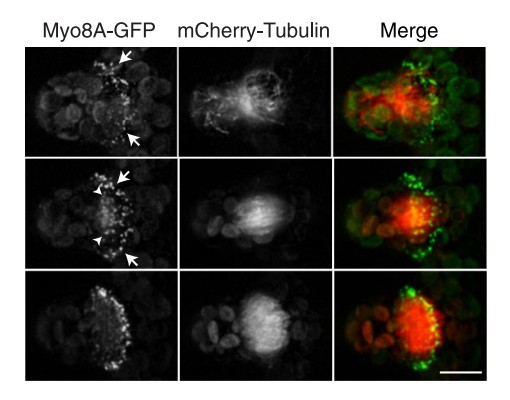

**Figure 3**. Myosin VIII localizes to the phragmoplast and cortical division site in moss. A protonemal branching cell expressing Myo8A-GFP (green) and mCherry-tubulin (red). Images are maximum intensity projections of z-stacks from a spinning disc time-series acquisition. Before mitosis, Myo8A-GFP accumulates at the neck of the emerging cell (top, arrows). Myo8A-GFP accumulates at the spindle midzone (middle, arrow heads) and forms a ring at the edge of the phragmoplast that expands out to the cell cortex (bottom). Scale bar, 10 μm. See also **Video 3**.

The following figure supplement is available for figure 3:

**Figure supplement 1**. Myo8A-GFP localizes to the cortical division site and the phragmoplast in moss.

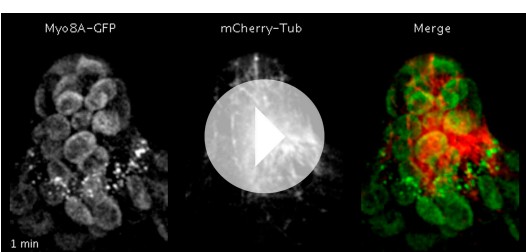

**Video 3**. Myo8A-GFP marks the future site of cell division in moss branching cells (see **Figure 3**). Moss branch cell expressing Myo8A-GFP (green) and mCherry-tubulin (red) was imaged with a spinning disc confocal microscope. Images are maximum projections of a z-stack acquired every minute. Video is playing at 4 fps. Scale bar, 10 μm.

presumptive position of the preprophase band (**Figure 4B**). As cells enter mitosis, cortical Myo8A-GFP tightens into a thin band and remains at the cortical division site. In early cytokinesis, Myo8A-GFP appears also at the phragmoplast midzone (**Figure 4C**). As the phragmoplast expands, the Myo8A-GFP at the phragmoplast midzone expands out eventually reaching the Myo8A-GFP at the cortical division site (**Figure 4D**; **Video 5**). Evidently, myo8A-GFP is capable of localizing to the phragmoplast and future site of division in a seed plant as well as in a bryophyte.

## Myo8A-GFP localizes to the spindle and phragmoplast independent of actin

Because division in apical cells is more frequent than in branching cells, we imaged moss apical cells to investigate the mechanism of myosin VIII function during phragmoplast expansion. Before nuclear envelope break down, we found a population of Myo8A-GFP that localizes to cytoplasmic microtubules surrounding the nucleus (**Figure 5A**). Myo8A-GFP remains localized along microtubules as the spindle is assembled (**Figure 5B**; **Video 6**). During mitosis, Myo8A-GFP continues to associate with the mitotic spindle (**Figure 5C**; **Video 7**), enriched at the midzone and to some extent at the poles. Prior to anaphase, a small population of Myo8A-GFP accumulates at the cortex near the spindle midzone (**Figure 5C**, arrows; **Video 7**). During anaphase, Myo8A-GFP concentrates at the midzone. Initially Myo8A-GFP is found throughout the phragmoplast midzone (**Figure 5C—figure supplement 1**). As the phragmoplast matures, Myo8A-GFP accumulates on the leading edge of the phragmoplast, ultimately forming a ring (**Figure 5C**, **Figure 5C—figure supplement 1**; **Video 7**). Interestingly in apical cells, Myo8A-GFP does accumulate at the cell cortex, but in contrast to branching cells, this accumulation occurs later and is significantly more dynamic.

To test whether actin is involved in recruiting Myo8A-GFP to the mitotic spindle, we first imaged Myo8A-GFP and lifeact-mCherry in dividing cells (**Figure 6A**; **Video 8**, top). When Myo8A-GFP appears on the spindle, there is little to no accumulation of lifeact-mCherry. During the transition from spindle to phragmoplast, Myo8A-GFP concentrates at the midzone and lifeact-mCherry fluorescence rises above background levels in the vicinity of the phragmoplast. However, when Myo8A-GFP fluorescence tightens into a thin band on the phragmoplast leading edge, the lifeact-mCherry fluorescence accumulates significantly around the phragmoplast (**Figure 6A**; **Video 8**, top). The timing of the appearance of actin suggests that early localization of Myo8A-GFP to the mitotic spindle is independent of actin. To test this, we imaged cells entering mitosis in the presence of 25 μM Latrunculin B, which depolymerizes the actin cytoskeleton (**Figure 6—figure supplement 1**). Strikingly, Myo8A-GFP

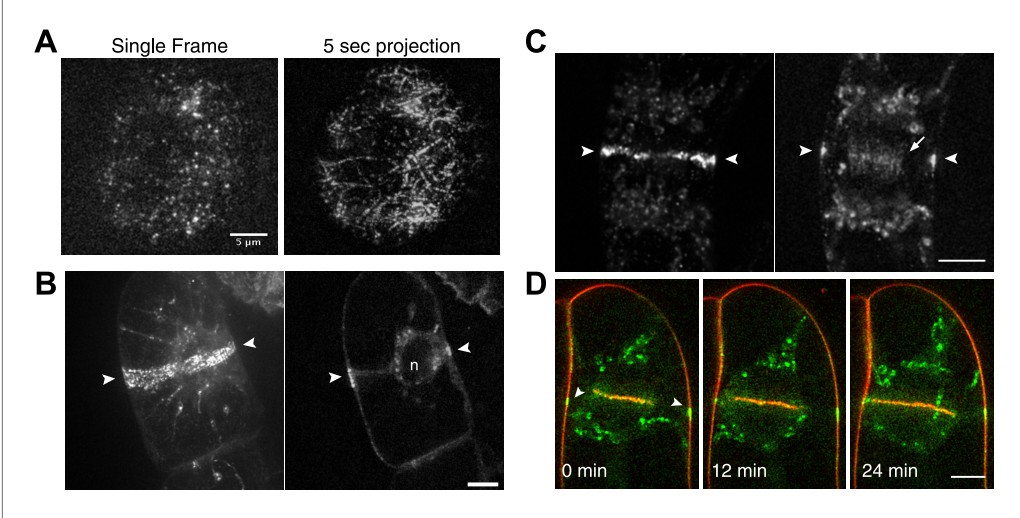

**Figure 4**. Myosin VIII localizes to the preprophase band, phragmoplast and cortical division site in tobacco BY-2 cells. (**A**) Myo8A-GFP localizes to dynamic punctate cortical structures on the cell cortex of BY-2 cells. VAEM image of a single frame from a time-lapse acquisition is shown on the left. On the right is a maximum projection of frames from 5 s of real time. Linear trajectories are readily apparent in the maximum projection. See also *Video 4*. Scale bar, 5 µm. (**B** and **C**) Left, z-projection. Right, midplane. Tobacco BY-2 cell in preprophase (**B**) and cytokinesis (**C**) expressing moss Myo8A-GFP. Myo8A-GFP accumulates on the preprophase band (**B**, arrow heads). n denotes the nucleus. Myo8A-GFP remains at the cortical division site (**C**, arrow heads) and is at the phragmoplast midzone (**C**, arrow). (**D**) Images from a single focal plane of a dividing BY-2 cell expressing Myo8A-GFP (green) and stained with FM4-64 (red) acquired over time. FM4-64 labels membrane added to the expanding cell plate (asterisk). Myo8A-GFP localizes to the phragmoplast midzone (arrow) and the cortical division site (arrow heads). See also *Video 5*. Images in (**B**–**D**) were acquired with a spinning disc confocal. (**B**–**D**) Scale bars, 10 µm.

The following source data are available for figure 4:

**Source data 1**. Multiple sequence alignment of class VIII myosins from *Arabidopsis thaliana* (At), *Nicotiana benthamiana* (Nb), and *Physcomitrella patens* (Pp) generated with Clustal O. *Nicotiana benthamiana* sequences can be found on CyMobase. The following sequences can be found on Phytozome *Physcomitrella patens* v1.6: PpMyo8A (Pp1s228_18V6.1), PpMyo8C (Pp1s199_21V6.1). The following sequences can be found on NCBI: PpMyo8B (AEM05967), PpMyo8D (AEM05968), PpMyo8E (AEM05969), AtMyo8A (NP_194467), AtMyo8B (NP_175453), AtMyo8C (NP_001078755), AtMyo8D (NP_188630).

**Source data 2**. Table shows the amino acid sequence comparison between *Arabidopsis thaliana* (At), *Nicotiana benthamiana* (Nb) and *Physcomitrella patens* (Pp) class VIII myosins. Percent identity from Clusal O multiple sequence alignment is reported.

still accumulates on the mitotic spindle. Similar to control cells, Myo8A-GFP in drug-treated cells accumulates in the spindle midzone at anaphase and tightens into a thin band at the leading edge of the phragmoplast during cytokinesis (*Figure 6B*; *Video 8*, bottom). Thus, Myo8A-GFP recruitment to the mitotic spindle and behavior during mitosis is apparently independent of actin.

## Myosin VIII function in cytokinesis requires actin

The fact that Myo8A-GFP localization is scarcely changed in the absence of actin raises the possibility that myosin VIII functions independently of actin. To test this, we imaged phragmoplasts labeled with GFP-tubulin and FM4-64. Since actin is essential for polarized expansion, it was not possible to perform long-term Latrunculin B treatments. Instead, plants were treated for 2 hr before imaging, ensuring that essentially all observed phragmoplasts had been formed in the absence of actin. The majority of untreated, wild-type phragmoplasts deposit membrane uniformly, appearing as smooth FM4-64 staining in the midzone (80% of wild type cells, n = 55; *Figure 7A*; *Video 9*,). In contrast, most myosin VIII null dividing cells have non-uniform FM4-64 staining (70% of Δmyo8ABCDE cells, n = 79; *Figure 7B*; *Video 10*). Interestingly, treatment of wild type dividing cells with Latrunculin B results in cells with similar

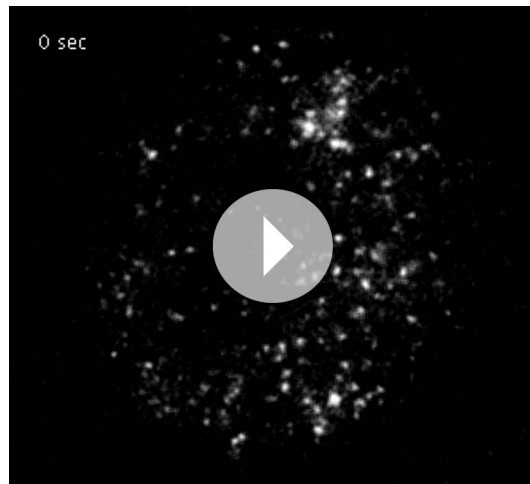

**Video 4**. Myo8A-GFP localizes to dynamic punctate cortical structures on the cell cortex of BY-2 cells (see **Figure 4A**). BY-2 cells expressing Myo8A-GFP were imaged with VAEM (acquired at 11.3 fps). Video is playing at 11.3 fps. Scale bar, 5 µm.

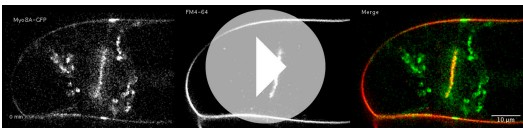

**Video 5**. Myo8A-GFP localizes to the phragmoplast and cortical division site in tobacco BY-2 cells (see **Figure 4D**). BY-2 cell expressing Myo8A-GFP (green) stained with FM4-64 (red) was imaged with spinning disc confocal microscope. Images are a single focal plane acquired every minute. Video is playing at 3 fps. Scale bar, 10 µm.

defects in cell plate assembly as those observed in the myosin VIII null cells (70% of wild type cells treated with LatB, n = 50; **Figure 7C**; **Video 11**), indicating that the observed changes in the assembling cell plate result from compromising actin and myosin.

During phragmoplast expansion, the vast majority of microtubules are tightly focused at the phragmoplast midzone, where there is a strong accumulation of Myo8A-GFP. However, we also observed that Myo8A-GFP localizes to the ends of peripheral phragmoplast microtubules (**Figure 8A,B**, arrows; **Video 12**) that initiate from the edge of the phragmoplast and are initially unattached to the phragmoplast midzone. We investigated the behavior of these peripheral microtubules in control and latrunculin B treated cells (**Figure 8**; **Video 12**). Before the phragmoplast reaches the cell cortex, we found that there were three times fewer peripheral microtubules in control cells as compared to cells treated with latrunculin B. We suspected that this difference likely results from the fact that peripheral microtubules more rapidly focus at the phragmoplast midzone in control cells. To test this, we identified peripheral microtubules in control and latrunculin B treated cells and followed them for 20 s. We found in control cells that only 20% of the peripheral microtubules were still present after 20 s (n = 4 cells). In contrast, 89% of peripheral microtubules were still present after 20 s in latrunculin B treated cells (n = 4 cells). In some cases (**Figure 8D**, yellow arrow) peripheral microtubules were observed to remain unattached for more than a minute. Taken together these data indicate that in the presence of actin peripheral microtubules are swiftly integrated into the expanding phragmoplast, suggesting that actin filaments exist between the edge of the expanding phragmoplast, peripheral microtubules, and the cell cortex.

In support of this, we found that an actin nucleator, For2A-GFP (**van Gisbergen et al., 2012**), is enriched on the phragmoplast as soon as it forms from the late spindle (**Figure 9A**; **Video 13**). For2A-GFP remains on the edge of the phragmoplast throughout cytokinesis (**Figure 9B**; **Video 14**), suggesting that actin is actively polymerized on the phragmoplast edge. To test this, we simultaneously imaged lifeact-mEGFP and mCherry-tubulin using a laser scanning confocal microscope. We confirmed that actin accumulates at the midzone once the phragmoplast forms. As the phragmoplast expands out from the center of the cell towards the cell cortex, we discovered that actin filaments are present at the midzone and between the leading edge of the phragmoplast and the cell cortex (**Figure 9C**; **Video 15**). Moreover, peripheral phragmoplast microtubules intersect the actin filaments that span the distance between the phragmoplast leading edge and the cell cortex (**Figure 9C**, arrowhead). Our data suggest that microtubules may interact with actin filaments bridging the cell cortex and the phragmoplast during the time that myosin VIII-mediated motility guides the phragmoplast.

## Discussion

Based on our data, we propose the following model for myosin VIII function in phragmoplast guidance during division of a protonemal apical cell (**Figure 10**). A population of myosin VIII localizes to microtubules and this population incorporates into the mitotic spindle upon spindle formation. In early

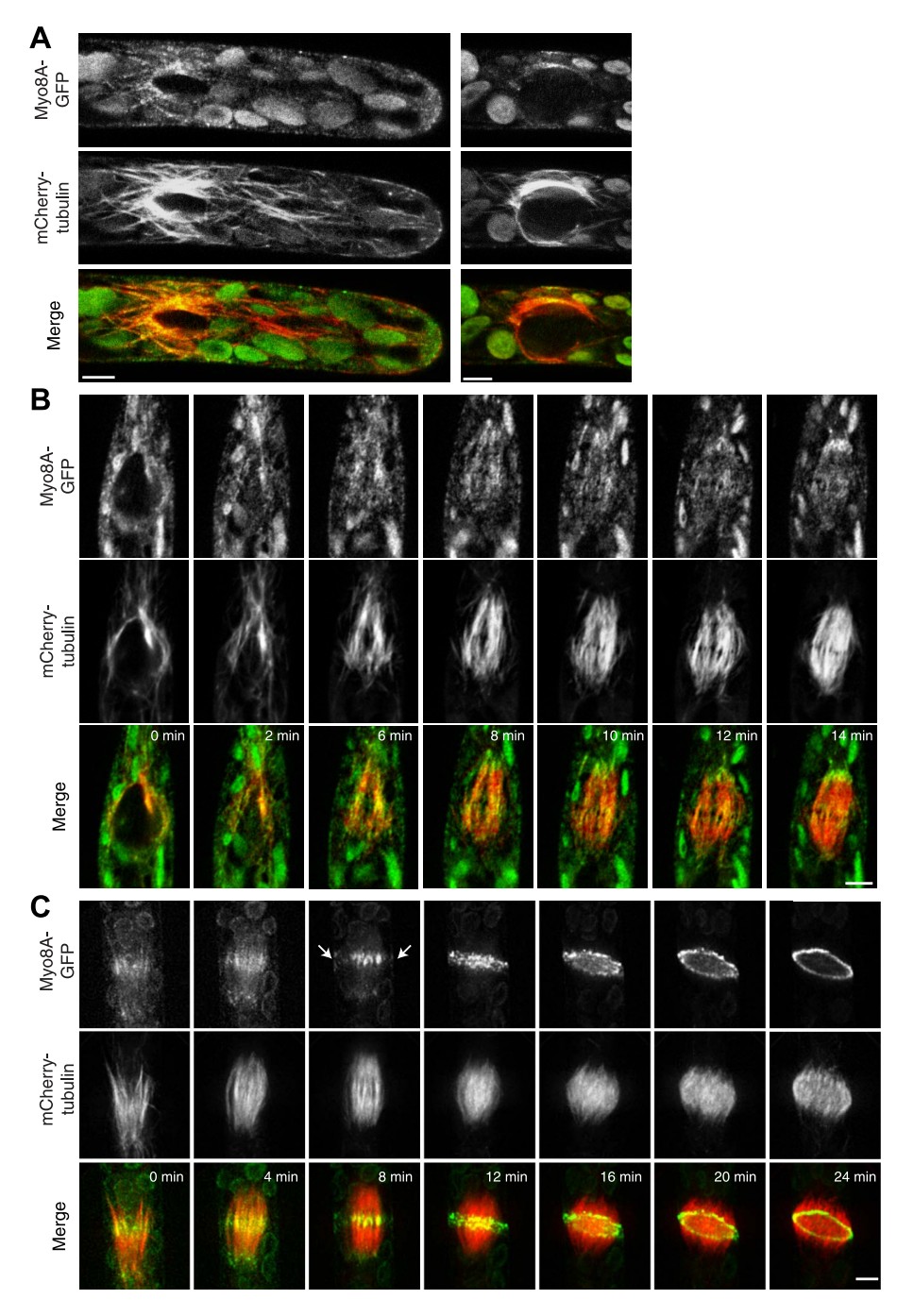

**Figure 5**. Myo8A-GFP localizes to the mitotic spindle and phragmoplast. (**A** and **B**) Moss protonemal apical cells expressing Myo8A-GFP (green) and mCherry-tubulin (red) imaged on a scanning confocal microscope. Images are single focal planes acquired over time. Scale bar, 5 µm. (**A**) Two examples of Myo8A-GFP associating with cytoplasmic microtubules surrounding the nucleus before mitosis. (**B**) Myo8A-GFP stays associated with microtubules throughout mitosis. See also *Video 6*. Scale bar, 5 µm. (**C**) Myo8A-GFP accumulates in the midzone. Arrows indicate cortical accumulation. Images were acquired with a spinning disc confocal microscope and are maximum projections of z-stacks acquired over time. See also *Video 7*. Scale bar, 5 µm. In all cases, large globular structures are chloroplasts that auto-fluoresce in the GFP channel.

The following figure supplement is available for figure 5:

**Figure supplement 1**. Myo8A-GFP localization in the phragmoplast.

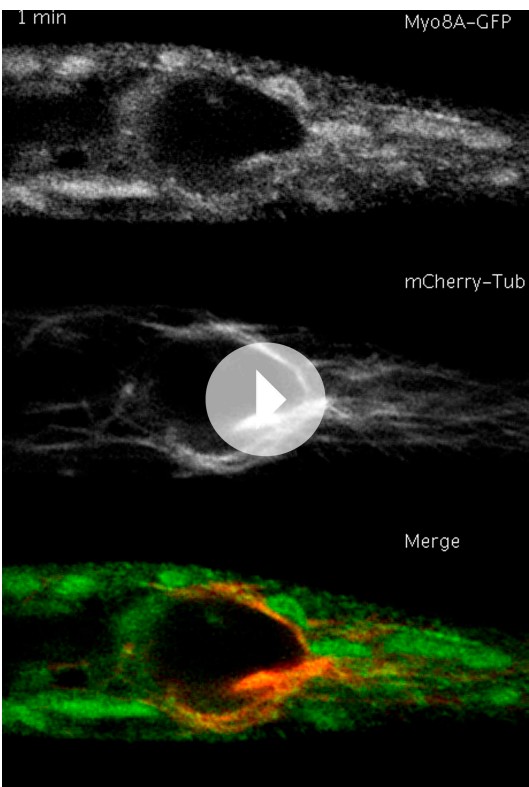

**Video 6**. Myo8A-GFP localizes to the cytoplasmic microtubules around nucleus and remains on the spindle (see **Figure 5B**). Moss apical cell expressing Myo8A-GFP (green) and mCherry-tubulin (red) was imaged with a scanning confocal microscope. Images are a single focal plane acquired every minute. Video is playing at 4 fps. Scale bar, 5 μm.

**Video 7**. Myo8A-GFP associates with mitotic spindle and phragmoplast (see **Figure 5C**). Moss apical cell expressing Myo8A-GFP (green) and mCherry-tubulin (red) was imaged with a spinning disc confocal microscope. Images are maximum projections of a z-stack acquired every minute. Video is playing at 4 fps. Scale bar, 5 μm.

mitosis, as microtubules are dynamically searching to make chromosomal attachments, Myo8A-GFP decorates the entire spindle (**Figure 5B**). At metaphase, Myo8A-GFP concentrates in the spindle midzone and at anaphase an additional population of Myo8A-GFP dynamically concentrates at the cell cortex where the cell plate ultimately fuses with the parental cell wall. Once the phragmoplast assembles, Myo8A-GFP forms a tight ring at the edge of the expanding phragmoplast (**Figure 5**). At this point, the class II formin, For2A, localizes to the phragmoplast (**Figure 9B**; **van Gisbergen et al., 2012**) and polymerizes actin filaments between the leading edge of the phragmoplast and the cell cortex (**Figure 9C**). Since For2A remains associated with the phragmoplast, it suggests that the barbed ends of the actin filaments are anchored at the phragmoplast midzone and the pointed ends are in the cytoplasm. Myosin VIII at the cortex can hold onto these actin filaments and walk towards the barbed end thereby aligning the actin filaments to the cortical division site. We propose that myosin VIII at the ends of peripheral phragmoplast microtubules moves along these actin filaments from the cortical division site towards the expanding phragmoplast, thereby translocating microtubules and ensuring that phragmoplast expansion occurs along a plane defined by the cortical division site (**Figure 10**).

There is precedent for myosin-based motility translocating microtubules on actin filaments. In the budding yeast, *Saccharomyces cerevisiae*, the class V myosin, Myo2p, binds Kar9p, which localizes to cytoplasmic microtubule plus ends by binding the yeast EB1 homolog, Bim1p (**Beach et al., 2000**). These cytoplasmic microtubules emanate from the spindle pole body embedded in the nuclear envelope and Myo2p mediates their motility along actin cables directed into the bud, moving the nucleus toward the bud neck (**Beach et al., 2000**; **Yin et al., 2000**). We imagine a similar mechanism could be at work in plant cells, whereby myosin VIII associates with microtubule ends, and subsequently translocates microtubules on actin filaments to guide phragmoplast expansion.

While the localization of myosin VIII is striking, myosin VIII in protonemata is mostly dispensable for phragmoplast guidance in apical cells. These cells are narrow, with diameters not much greater than the sizes of the nucleus and spindle, leaving little room for those two structures to be mis-placed. Due to these geometric constraints, the mitotic spindle always forms along the long axis of the cell. Thus, any cell plate positioning defects that occur are mild because the spindle midzone is always roughly perpendicular to the long axis of the cell.

In contrast, myosin VIII is needed for branch formation, insofar as cell plates are often aberrantly positioned in the branching cells of the myosin VIII nulls. Arguably, in comparison to apical cell

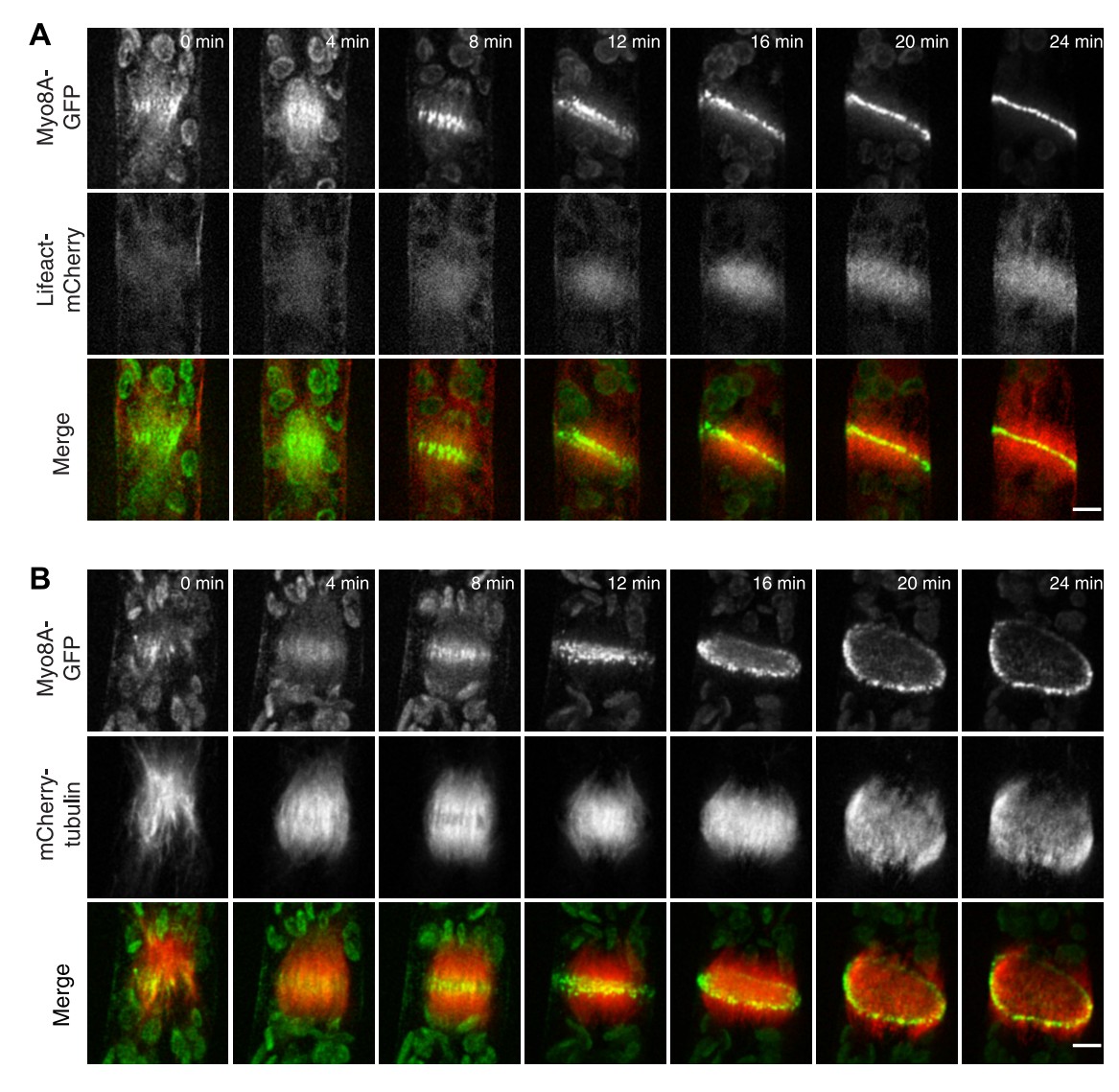

**Figure 6**. Myo8A-GFP localizes to the mitotic spindle and phragmoplast independent of actin. Moss protonemal apical cells imaged on a spinning disc confocal microscope. All images are maximum projections of z-stacks acquired over time. (**A**) Cell expressing Myo8A-GFP (green) and Lifeact-mCherry (red). (**B**) Cell expressing Myo8A-GFP (green) and mCherry-tubulin (red) treated with 25 μm LatB. See also *Video 8*. Scale bars, 5 μm. In all cases, large globular structures are chloroplasts that auto-fluoresce in the GFP channel.

The following figure supplement is available for figure 6:

**Figure supplement 1**. Dose response of latrunculin B in apical protonemal moss cells.

division, side-branch formation needs to specify the cell division plane more accurately. Branching involves an asymmetric cell division in an L-shaped cell. The nucleus migrates toward the junction of the parental cell and the branch and the spindle is oriented along the longitudinal axis of the new emerging cell. The phragmoplast builds the new cell plate at the junction of the two cells. We found that in branching cells, a relatively static population of Myo8A-GFP accumulates at the cell cortex at the future site of cell division (*Figure 3*). Myo8A-GFP accumulates at this site early, prior to mitosis, and remains throughout cytokinesis. As in apical cells, Myo8A-GFP also accumulates on the mitotic spindle, ultimately forming a tight band on the leading edge of the phragmoplast. The new cell plate fuses with the parental cell membranes at the cortical site defined by the presence of Myo8A-GFP. Since there are many ways to orient the spindle in a branching cell, in the absence of myosin VIII

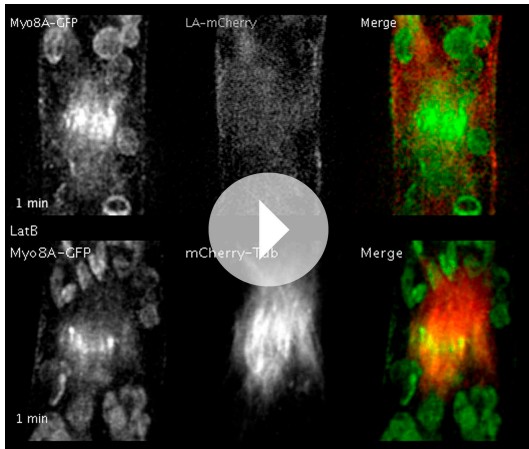

**Video 8**. Myo8A-GFP associates with the mitotic spindle and phragmoplast independent of actin (see *Figure 6A,B*). Moss apical cells expressing Myo8A-GFP (green) and lifeact-mCherry/mCherry tubulin (red) were imaged with a spinning disc confocal microscope. Images are maximum projections of a z-stack acquired every minute. Video is playing at 4 fps. Scale bar, 5 µm. Top, Myo8A-GFP and lifeact-mCherry in control cell. Bottom, Myo8A-GFP and mCherry-tubulin in LatB treated cell.

defects in cell plate positioning are frequent, suggesting that myosin VIII functions to guide phragmoplast expansion.

Notably, our data suggests that a myosin VIII-mediated phragmoplast-guidance mechanism also exists in seed plants, since we found that moss Myo8A-GFP localizes to the preprophase band, cortical division site, and the phragmoplast midzone in BY-2 cells (*Figure 4*). Our result is consistent with a previous report in which a GFP fusion of one of the Arabidopsis class VIII myosins, ATM1-GFP, was shown to localize to the phragmoplast in BY-2 cells (*Van Damme et al., 2004*). The discovery that Myo8A-GFP localizes to the cortical division site gives us a portal to connect our observations in moss to the current model of cell division in seed plants (*Van Damme, 2009*; *Rasmussen et al., 2011b*). In that model, cells with preprophase bands use a microtubule-dependent mechanism to position the nucleus such that it is bisected by the future plane of division and to form the spindle perpendicularly (*Mineyuki and Furuya, 1986*; *Venverloo and Libbenga, 1987*; *Katsuta et al., 1990*). Once the spindle is aligned, defects in phragmoplast guidance would little alter cell plate positioning and subsequent tissue morphogenesis, obscuring the role of myosin VIII. Nevertheless, when BY-2 cells are treated with actin inhibitors, phragmoplasts are disorganized, generating wrinkled cell plates that are often skewed with respect to the cortical division site (*Hoshino et al., 2003*; *Yoneda et al., 2004*; *Sano et al., 2005*; *Higaki et al., 2008*; *Kojo et al., 2013*). Thus, we predict that for fine tuning myosin VIII and actin guide phragmoplast expansion in cells with preprophase bands.

Our data provide evidence that myosin VIII and actin steer phragmoplast expansion during cytokinesis in both moss and tobacco, suggesting that myosin VIII function is conserved throughout plant evolution. In fact, myosin VIII provides a physical link between phragmoplast microtubules and the cortical division site via actin filaments. We propose that myosin VIII's motor activity along actin provides a molecular mechanism for steering phragmoplast expansion.

## Materials and methods

### Plasmid construction

All expression constructs were constructed using Multisite Gateway recombination technology from Invitrogen (Carlsbad, CA). Generation of entry clones 3XmEGFP-L5L2 (*29*), Lifeact-L1R5 (*30*) and mCherry-L5L2 (*15*) were described previously. To construct the entry clone Myo8A-L1R5, total RNA was extracted from 7-day-old moss protonemal tissues using RNeasy plant mini kit (Qiagen), followed by DNase I treatment according to the manufacturer's protocol. cDNA was synthesized from total RNA using SuperScript II reverse transcriptase (Invitrogen) and oligo (dT) according to manufacturer's protocol. Full length Myo8A coding sequence was amplified from moss cDNA using Myo8A specific primers (P1 & P2), and cloned into pGEM-T easy from Promega (Madison, WI). The full-length Myo8A coding sequence was then amplified from the pGEM-Myo8A clone using primers (P3 & P4) containing attB1 and attB5r sites, and cloned into pDONR-P1P5r with a BP reaction (Invitrogen). The mCherry and mEGFP coding sequence was amplified using primers (P5 & P6) with attB1 and attB5r sites and cloned into pDONR-P1P5r to generate entry clones mCherry-L1R5 and mEGFP-L1R5. Moss α−tubulin coding sequence was amplified from pAct-GFP-TUA1 (*31*) with primers (P7 & P8) containing attB5 and attB2 sites and cloned into pDONR-P5P2 to generate entry clone α-tubulin-L5L2. Combinations of entry clones were assembled with destination vectors generating constructs for stable expression in

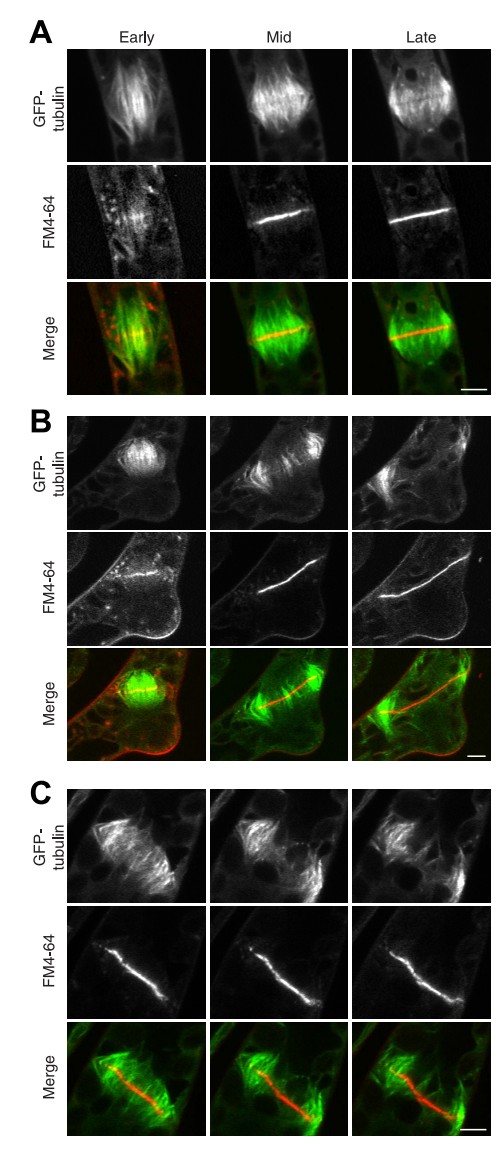

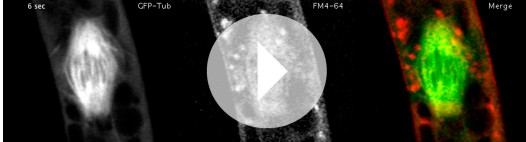

**Video 9**. New membrane is deposited uniformly in a wild type dividing cell (see *Figure 7A*). A wild type cell expressing GFP-tubulin was stained with FM4-64 and imaged in a single focal plane on a scanning confocal microscope. Video is playing at 10 fps. Scale bar, 5 μm.

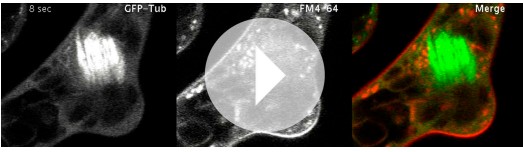

**Video 10**. New membrane is deposited non-uniformly in a Δmyo8ABCDE dividing cell (see *Figure 7B*). A Δmyo8ABCDE cell expressing GFP-tubulin was stained with FM4-64 and imaged in a single focal plane on a scanning confocal microscope. Video is playing at 10 fps. Scale bar, 5 μm.

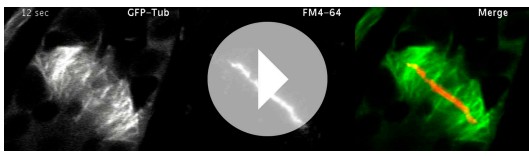

**Video 11**. New membrane is deposited non-uniformly in a wild type dividing cell treated with LatB (see *Figure 7C*). A wild type cell expressing GFP-tubulin was treated for two hours with 25 μM LatB and then stained with FM4-64 and imaged in 25 μM LatB. Images are from a single focal plane taken on a scanning confocal microscope. Video is playing at 10 fps. Scale bar, 5 μm.

**Figure 7**. Actin is required for Myo8 function in cytokinesis. Phragmoplasts from wild type. (**A**) Δmyo8ABCDE (**B**) and wild type treated with 25 μM LatB (**C**) expressing GFP-tubulin (green) and stained with FM4-64 (red). Cells were imaged on a scanning confocal microscope. For LatB treatment, wild type plants were treated with 25 μM LatB for 2 hr, then stained with FM4-64 and imaged in the presence of 25 μM LatB. Images are single focal planes taken from a time series. Scale bars, 5 μm. See also *Video 9* (for A), *Video 10* (for B), and *Video 11* (for C).

moss or tobacco BY-2 cells using LR clonase II plus reactions (Invitrogen) as follows: Myo8A-L1R5 and 3XmEGFP-L5L2 with pTKUbi-gate generating pTKUbi-Myo8A-3mEGFP; Lifeact-L1R5 and mCherry-L5L2 with pTZUbi-gate generating pTZUbi-Lifeact-mCherry; TUA1-L5L2 and mCherry-L1R5 with pTZUbi-gate generating pTZUbi-mCherry-tubulin; TUA1-L5L2 and mEGFP-L1R5 with pTKUbi-gate generating pTKUbi-mEGFP-tubulin; Myo8A-L1R5 and 3XmEGFP-L5L2 with pMDC32 (*32*) generating pMDC32-Myo8A-3XmEGFP.

The pTKUbi-gate vector has an expression cassette derived from pTHUbi-Gate(*33*), which contains the maize ubiquitin promoter, Gateway cassette and NOS terminator. Following this expression cassette is a 35S::NptII::ter cassette flanked by lox sites(*14*). The expression and antibiotic resistance

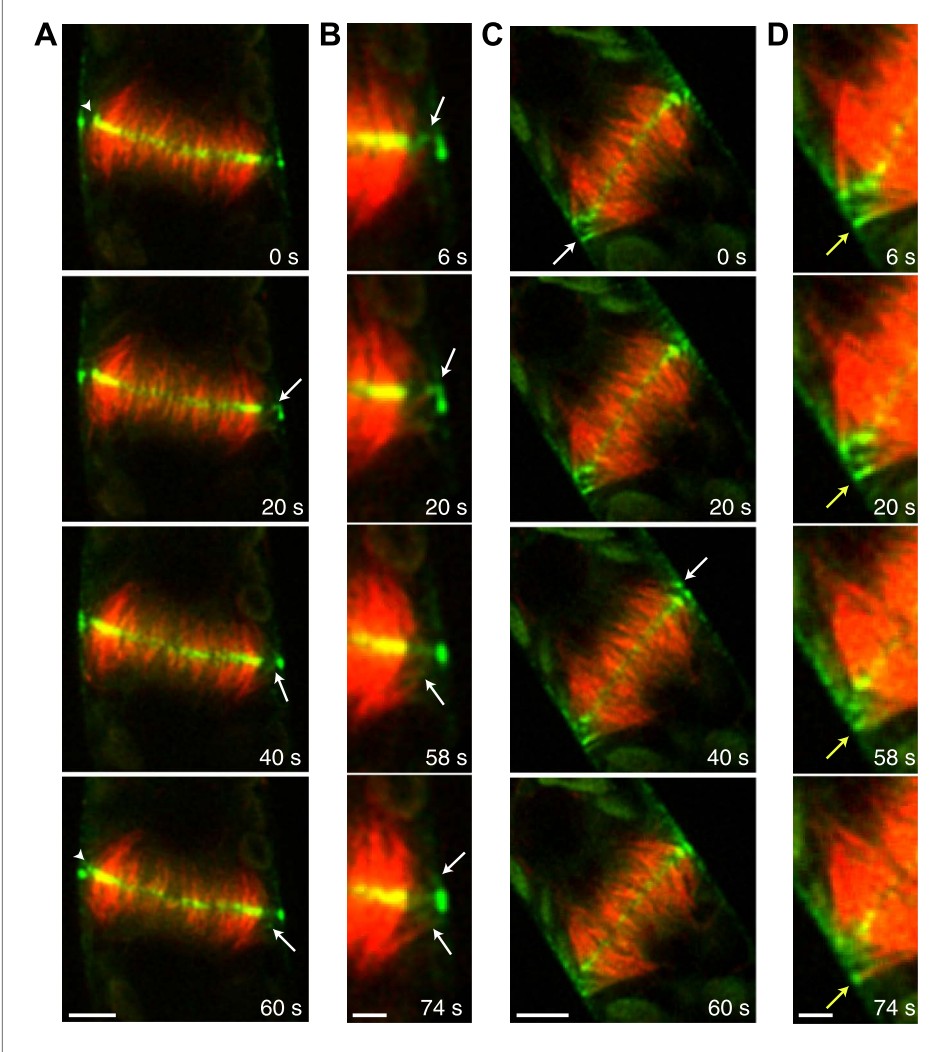

**Figure 8**. Myo8A-GFP associates with the ends of phragmoplast microtubules. Images of a protonemal apical cell expressing Myo8A-GFP (green) and mCherry-tubulin (red) were acquired with a spinning disc confocal microscope. Images are a single focal plane from a time series. Arrows indicate enrichment of Myo8A-GFP at the ends of peripheral microtubules. Arrow heads indicate where peripheral microtubules are incorporated into the phragmoplast midzone. (**A**) In a control cell, peripheral microtubules focus at the phragmoplast midzone. Scale bar, 5 μm. (**B**) Zoom-in of the phragmoplast periphery from the control cell. Peripheral microtubules with Myo8A-GFP are evident in this area. In the presence of actin, peripheral microtubules are incorporated into the phragmoplast midzone rapidly. Scale bar, 2 μm. (**C**) In a cell treated with 25 μM LatB, peripheral microtubules are no longer focused at the midzone. Scale bar, 5 μm. (**D**) Zoom-in of the phragmoplast periphery in the LatB treated cell. Peripheral microtubules stay associated with the cell cortex for more than a minute. Scale bar, 2 μm. See also **Video 12**.

cassettes are flanked by moss genomic sequence from the Pp1s249_67V6.1 locus. Nucleotides −2 to −1153 and nucleotides 660 to 1757 are on the 5′ and 3′ ends, respectively, with Pme I sites incorporated such that digestion with Pme I releases the moss genomic DNA targeting arms as well as the expression and resistance cassettes. The pTZUbi-gate vector is similar except that it contains a 35S::Zeo::ter cassette flanked by lox sites (*14*) as the antibiotic resistance cassette, uses moss genomic sequence from the Pp1s141_25V6.1 locus (nucleotides +908 to +2021 and −35 to −1532), and has Swa I for release of the moss genomic DNA targeting arms and expression and resistance cassettes.

The coding sequence of moss α-tubulin (Pp1s215_51V6 locus) was amplified from moss cDNA using primers P9 and P10 and cloned into pENTR/D-TOPO (Invitrogen). After verification by sequencing, the coding sequence was cloned into the L5L4-mCherry plasmid (modified with restriction sites C-terminal

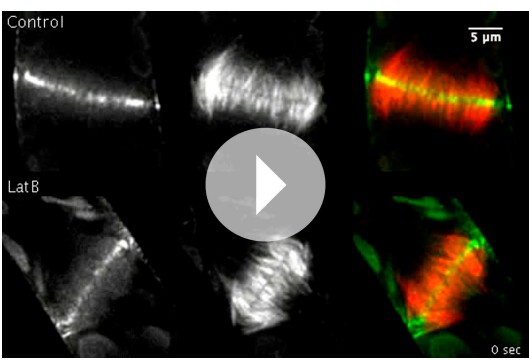

**Video 12**. Peripheral microtubules require actin to be efficiently incorporated into the phragmoplast (see **Figure 8**). Moss apical cell expressing Myo8A-GFP (green) and mCherry-tubulin (red) were imaged with a spinning disc confocal microscope. Images are a single focal plane acquired every 2 s. Video is playing at 10 fps. Scale bar, 5 µm. Top, control. Bottom, LatB treated cell.

to the mCherry L5L4-mCherry-AscI-SpeI) using the restriction sites AscI and SpeI, generating L5L4-mCherry-αtub215-51-1. Sequences upstream and downstream of the locus were amplified as targeting sequences for homologous recombination. The 5' targeting sequence was amplified using primers (P11 & P12) and cloned into pDONR-P1P5r using BP Clonase II (Invitrogen), generating L1R5-αtub215-51-1-5'tarm. Similarly, the 3' targeting sequence was amplified with primers (P13 & P14) and cloned into pDONR-P3P2 using BP Clonase II (Invitrogen), generating L3L2-αtub215-51-1-3'tarm. The L1R5-αtub215-51-1-5'tarm, L5L4-mCherry-αtub215-51-1 and L3L2-αtub215-51-1-3'tarm were recombined with R4R3 NOSter-Lox-Hygro-Lox (*34*) and pGEM-Gate (*30*) using LR Clonase II plus (Invitrogen) to generate the final construct for homologous recombination in moss, mCherry-αtub215-51-1AR.

## Primers used in this study

(P1) ATGTATTCTACGAATGGCATTGAGG; (P2) CTAACCTTGGAGCGCTCTTGAGG; (P3) GGGGAC AAGTTTGTACAAAAAAGCAGGCTTCATGTATTCTACGAATGGC; (P4) GGGGACAACTTTTGTATACAAA GTTGTACCTTGGAGCGCTCTTGAGG; (P5) GGGGACAAGTTTGTACAAAAAAGCAGGCTTCATGGT GAGCAAGGGCGAGGAG; (P6) GGGGACAACTTTTGTATACAAAGTTGTCTTGTACAGCTCGTCCA TGCC; (P7) GGGGACAACTTTTGTATACAAAGTTGTTATGAGAGAGATTATCAGCATCCAC; (P8) GGGG ACCACTTTGTACAAGAAAGCTGGGTATCAGTAGTCGTCGTCCTCC; (P9) ACTAGTTTAGTACTCGTC GTCGTCCTGTCCTCCGTCGGTGGATTCAGC; (P10) GGCGCGCCATGAGAGAGATCATCAGTATCCATATA GGTCAGG; (P11) GGGGACAAGTTTGTACAAAAAAGCAGGCTCTGGCGCGCCACTTCATAATCTACCT GTGC; (P12) GGGGACAACTTTTGTATACAAAGTTGTGGAAGAGTACGAGCAGCAGC; (P13) GGGG ACAACTTTGTATAATAAAGTTGTGGGCTTTTATTTTGAGGCGGAAACGG; (P14) GGGGACCACTTTGTA CAAGAAAGCTGGGTAGGCGCGCCGTTAACTGTGGAGTTCTG; (P15) GCAATACAACACACTGTGCTT GGG; (P16) GGTGTTGAAAGCATCATCACCACC.

## Plant materials and growth conditions

All moss tissue culture, protoplasting and transformation were performed as described previously (*15*). Moss protonemal tissues were propagated weekly on PpNH4 medium (1.03 mM MgSO$_4$, 1.86 mM KH$_2$PO$_4$, 3.3 mM Ca(NO$_3$)$_2$, 2.7 mM (NH$_4$)$_2$-tartrate, 45 µM FeSO$_4$, 9.93 µM H$_3$BO$_3$, 220 nM CuSO$_4$, 1.966 µM MnCl$_2$, 231 nM CoCl$_2$, 191 nM ZnSO$_4$, 169 nM KI, and 103 nM Na$_2$MoO$_4$) containing 0.7% agar. For imaging, 1-week-old protonemal tissues were protoplasted, plated in plating medium at a density of ~20,000 cells/9 cm$^2$ plate, regenerated on protoplast regeneration medium (PRM) for 4 days and transferred to PpNH4 plates. For moss transformation, protoplasts were isolated from 7-day-old moss protonemal tissue, and transformed with linearized DNA via PEG-mediated transformation (*15*). At least 30 µg of plasmid DNA was linearized, ethanol precipitated, and dissolved in sterile TE buffer. pTKUbi constructs were linearized with Pme I and pTZUbi constructs were linearized with Swa I. Protoplasts were transformed with linearized DNA via PEG-mediated transformation (*15*). Protoplasts were plated in top agar, regenerated on PRM (PpNH$_4$ medium supplemented with 8.5% mannitol and 10 mM CaCl$_2$.) for 4 days and transferred to PpNH$_4$ medium containing the appropriate antibiotics (G418, 20 µg/ml; zeocin, 50 µg/ml; hygromycin, 15 µg/ml). To select for stable transformants, transformations were cycled on and off antibiotic selection for three 1-week intervals. Stable transgenic lines were visually screened on a confocal microscope for expression of the transgene.

pTKUbi-mEGFP-Tub was transformed into WT and Δmyo8ABCDE (*14*) generating the mEGFP-Tub lines. pTKUbi-Myo8A-3mEGFP was transformed into Δmyo8ABCDE generating the Myo8A-GFP line. Myo8A-GFP was subsequently transformed with pTZUbi-mCherry-tubulin and pTZUbi-Lifeact-mCherry to generate Myo8A-GFP/mCherry-tub and Myo8A-GFP/Lifeact-mCherry.

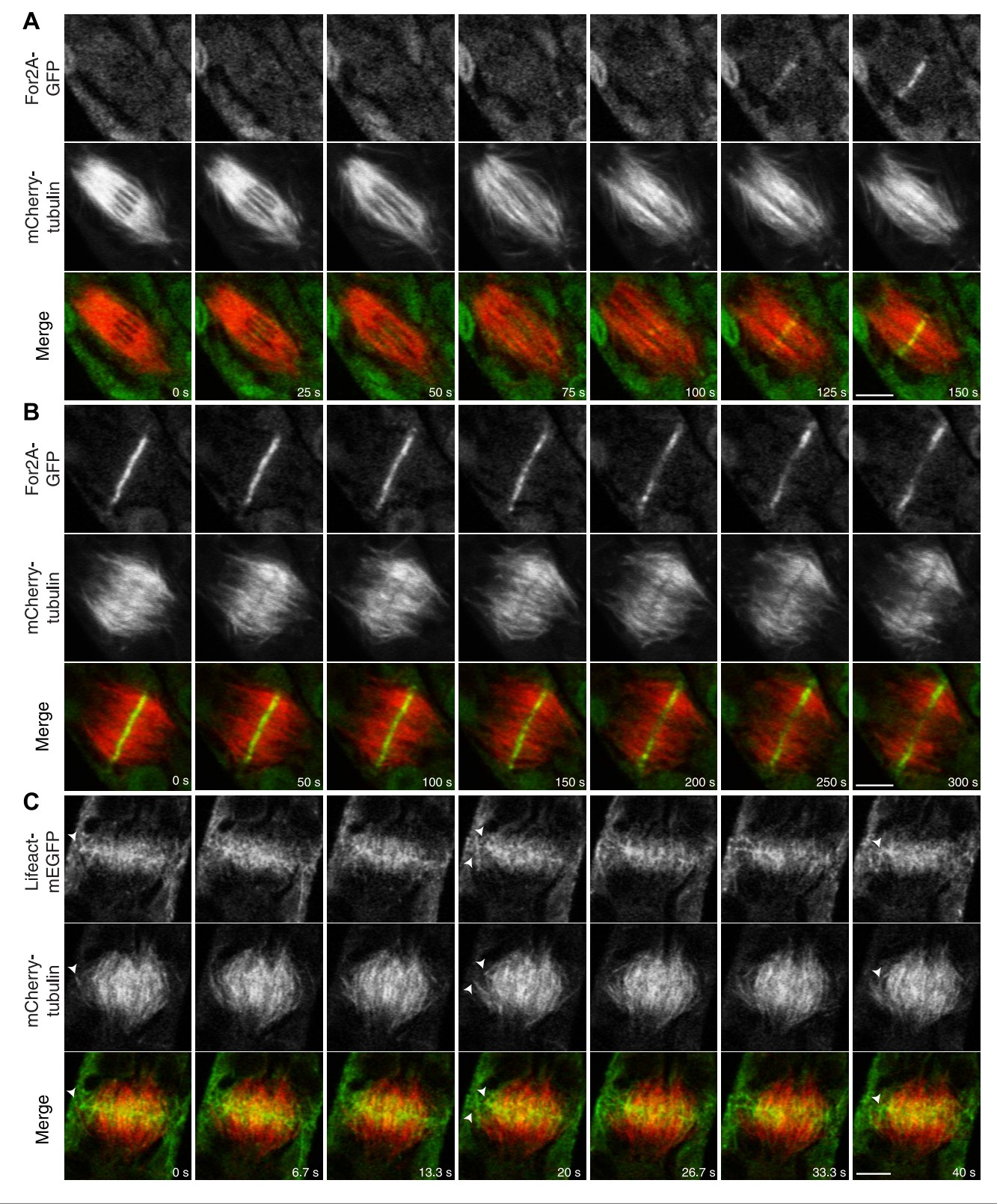

**Figure 9**. Actin is polymerized on the phragmoplast edge. Single focal plane images acquired with a laser scanning confocal microscope. (**A** and **B**) Protonemal apical cell expressing For2A-GFP (green) and mCherry-tubulin (red). (**A**) In metaphase through anaphase (**A**, t = 0–75 s), For2A-GFP is not associated with the spindle. See also **Video 13**. (**B**) For2A-GFP is enriched at the phragmoplast midzone and remains on the edge of the phragmoplast

*Figure 9. Continued on next page*

*Figure 9. Continued*

throughout cytokinesis. See also ***Video 14***. (**C**) Protonemal cell expressing Lifeact-mEGFP (green) and mCherry-tubulin (red). Microtubules intersect actin filaments between the leading edge of the phragmoplast and the cell cortex (arrow heads). See also ***Video 15***. Scale bars, 5 μm.

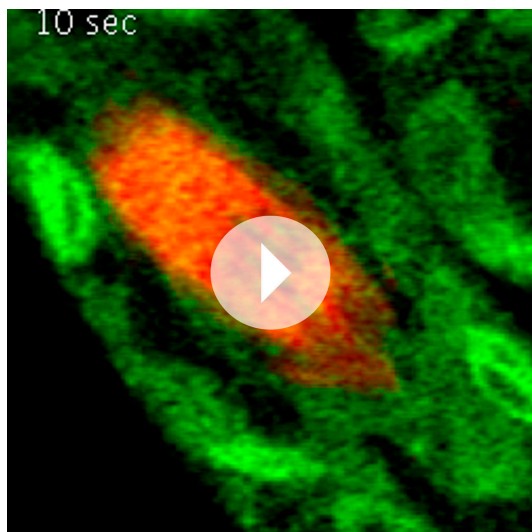

**Video 13**. For2A-GFP does not associat with the mitotic spindle but is present in the phragmoplast (see ***Figure 9A***). Moss apical cell expressing For2A-GFP (green) and mCherry-tubulin (red) was imaged with a scanning confocal microscope. Images are a single focal plane acquired every 5 s. Video is playing at 5 fps. Scale bar, 5 μm.

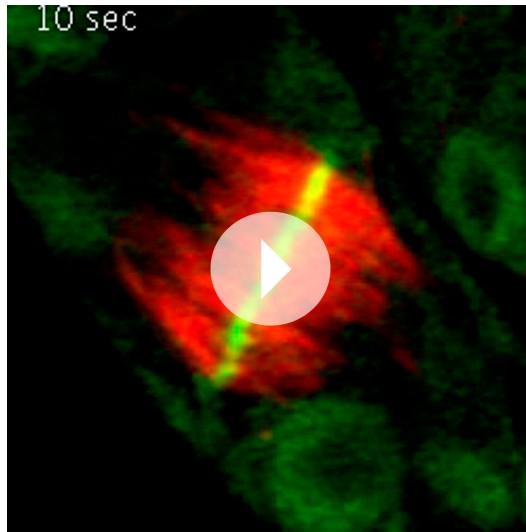

**Video 14**. For2A-GFP remains at the edge of phragmoplast (see ***Figure 9B***). Moss apical cell expressing For2A-GFP (green) and mCherry-tubulin (red) was imaged with a scanning confocal microscope. Images are a single focal plane acquired every 5 s. Video is playing at 5 fps. Scale bar, 5 μm.

Lifeact-mEGFP line (*30*) was transformed with the Asc I linearized mCherry-αtub215-51-1AR construct, to generate Lifeact-mEGFP/mCherry–tubulin line. Stably transformed lines were identified by visually screening for mCherry expression and then verified by genotyping using primers P15 and P16.

Tobacco BY-2 tissue culture and Agrobacterium-mediated transformation were performed as previously described (*35*). Tobacco BY-2 cells were transformed with pMDC32-Myo8A-3XmEGFP. The GV3101 *Agrobacterium tumefaciens* strain was used for infection.

## Phenotypic analysis, cell wall staining, and imaging

For measurement of cell plate angle and apical cell length, plants regenerated from protoplasts were stained with 0.1 mg ml$^{-1}$ calcofluor solution and visualized by epifluorescence microscopy (Leica MZ16FA, Leica Microsystems, Buffalo Grove, IL) using a UV filter (excitation 360/40 nm, emission 420 long pass) or a Violet filter (excitation 425/40 nm, emission 400 long pass). Cell plate angles and apical cell lengths were measured manually using ImageJ software.

## Sample preparation for confocal and VAEM imaging

For imaging cell division, 5- to 8-day-old plants regenerated from protoplasts were placed unto an agar pad in Hoagland's (4 mM $KNO_3$, 2 mM $KH_2PO_4$, 1 mM $Ca(NO_3)_2$, 89 μM Fe citrate, 300 μM $MgSO_4$, 9.93 μM $H_3BO_3$, 220 nM $CuSO_4$, 1.966 μM $MnCl_2$, 231 nM $CoCl_2$, 191 nM $ZnSO_4$, 169 nM KI, 103 nM $Na_2MoO_4$, and 1% sucrose), covered by a glass cover slip and sealed with VALAP (1:1:1 parts of Vaseline, lanoline, and paraffin). For cell plate staining, 15 μM FM4-64 was added in the Hoagland's solution in the agar pad. For latrunculin B treatment, plants were transferred to PpNH4 medium containing 25 μM latrunculin B for 2 hr, then transferred to slides containing 25 μM latrunculin B in both the agar pad and the Hoagland's solution.

## Variable angle epifluorescence microscopy (VAEM)

Samples were mounted on an inverted microscope (model Ti-E; Nikon Instruments, Melville, NY) equipped with a mirror-based T-FL-TIRF

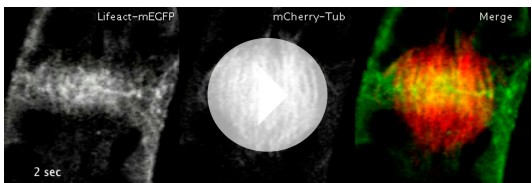

**Video 15**. Actin polymerizes between the edge of the phragmoplast and the cortical division site during cytokinesis (see *Figure 9C*). Moss apical cell expressing Myo8A-GFP (green) and lifeact-mCherry (red) was imaged with a scanning confocal microscope. Images are a single focal plane acquired continuously at 3.75 fps. Video is playing at 75 fps. Scale bar, 5 µm.

illuminator (Nikon) and imaged with a 1.49 NA 100× oil immersion TIRF objective (Nikon). The 1.5× optivar was used for all images to increase magnification. 488 and 561 nm laser illumination was used for GFP and mCherry excitation, respectively. The laser illumination angle was adjusted individually for each sample to achieve the maximum signal-to-noise ratio. Signals for each channel were captured simultaneously with a 1024 × 1024 electron-multiplying CCD camera (iXON3; Andor Technology USA, South Windsor, CT) equipped with a dual-view adaptor (Photometrics, Tucson, AZ). Emission filters were 525/50 nm for GFP and 595/50 for mCherry. Image acquisition process was controlled by NIS-Elements AR 3.2 software (Nikon).

### Spinning-disc confocal imaging

The slides were mounted on an inverted microscope (model Ti-E; Nikon) equipped with a Yokogawa spinning disk head (model CSU-X1) and a 512 × 512 electron multiplying CCD camera (iXON; Andor

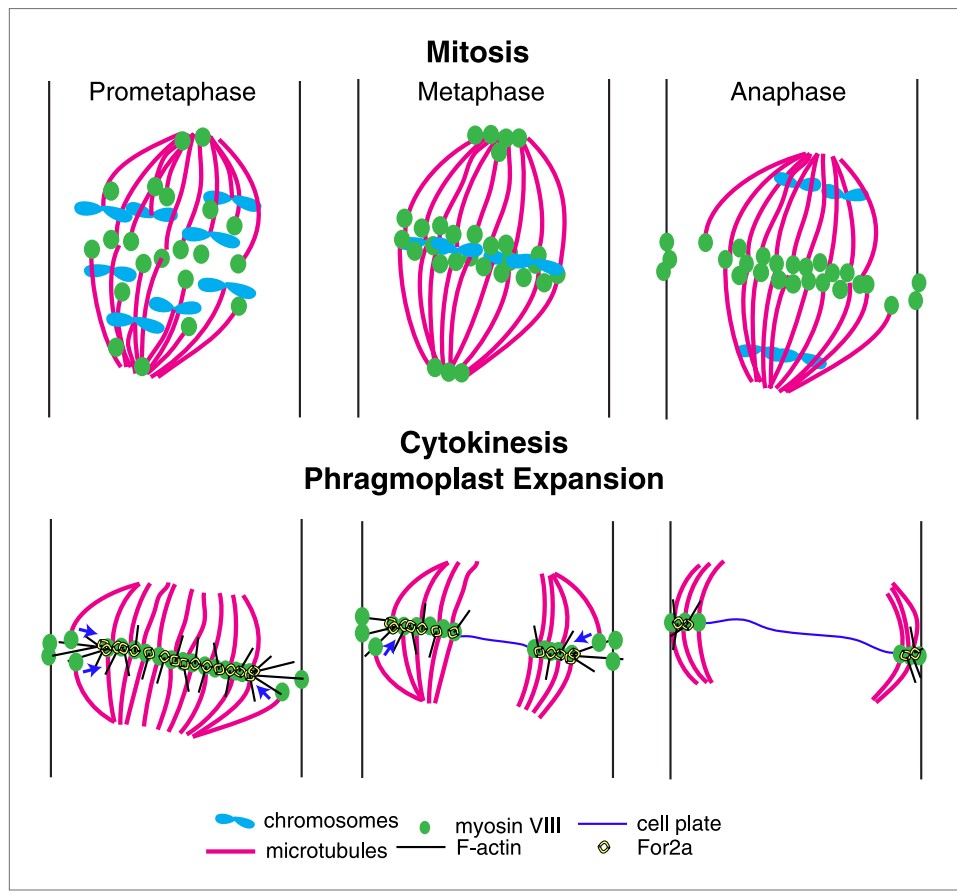

**Figure 10**. A model for myosin VIII function in phragmoplast guidance. In prometaphase, myosin VIII localizes to plus ends throughout the mitotic spindle. In metaphase, myosin VIII accumulates at the spindle midzone and on the poles. During anaphase, myosin VIII is observed on peripheral microtubules and at the cell cortex. As the phragmoplast forms, the midzone myosin VIII accumulation tightens into a thin band on the phragmoplast edge. For2A is at the phragmoplast midzone. Actin filaments are generated between the phragmoplast and cortical myosin VIII. Peripheral microtubules with plus-end associated myosin VIII translocate on actin filaments and are incorporated into the expanding phragmoplast.

Technology). Images for each channel were collected sequentially with a 1.4 NA 100× oil immersion objective (Nikon) at room temperature. 488 and 561 nm laser illumination was used for GFP/FM4-64 and mCherry excitation, respectively. Emission filters were 515/30 nm for GFP and 600/32 nm for mCherry/FM4-64. Image acquisition was controlled by MetaMorph software (Molecular Devices, Sunnyvale, CA).

### Laser scanning confocal microscopy

Images for each channel were acquired simultaneously on a Nikon A1R confocal microscope system with a 1.4 NA 100× oil immersion objective (Nikon) at room temperature. 488 and 561 nm laser illumination was used for GFP/FM4-64 and mCherry excitation, respectively. Emission filters were 525/50 nm for GFP and 595/50 for mCherry/FM4-64. Image acquisition was controlled by NIS-Elements software (Nikon).

### Image processing

All images were processed using ImageJ software with enhanced contrast and background subtraction (rolling ball diameter of 50 was used for spinning disc images; for scanning confocal images a rolling ball diameter of 50 was used for actin filaments and 75 for all other labels). We also performed smoothing and applied the unsharp mask filter to all images. All settings were standard.

## Acknowledgements

We thank Tobias Baskin for extensive editing of the manuscript. We thank Patricia Wadsworth, Peter Hepler, Heidi Rutschow, and Peter van Gisbergen for critical reading and helpful suggestions. We are particularly grateful to Graham Burkart and Alexis Tomazewski for generating the mCherry-tubulin/lifeact-mEGFP moss stable line. We thank Baskin lab members for technical advice regarding BY-2 cell culture and transformation.

## Additional information

### Funding

| Funder | Grant reference number | Author |
| --- | --- | --- |
| David and Lucile Packard Foundation | 2007-31750 | Magdalena Bezanilla |
| National Science Foundation | 0747231 | Magdalena Bezanilla |
| Marine Biological Laboratory | Nikon award | Magdalena Bezanilla |

The funders had no role in study design, data collection and interpretation, or the decision to submit the work for publication.

### Author contributions

S-ZW, MB, Conception and design, Acquisition of data, Analysis and interpretation of data, Drafting or revising the article

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
