## [Decision Letter]

Thank you for sending your work entitled “Myosin VIII associates with microtubule ends and together with actin steers plant cell division” for consideration at *eLife.* Your article has been favorably evaluated by Detlef Weigel (Senior editor) and 3 reviewers, one of whom is a member of our Board of Reviewing Editors.

The Reviewing editor and the other reviewers discussed their comments before we reached this decision, and the Reviewing editor has assembled the following comments to help you prepare a revised submission.

1) While the models presented in this manuscript are thought provoking, they are still quite speculative, and the text and title should be modified to reflect the fact that there is limited evidence for the guidance of the phragmoplast to the cortical division site. The only result that directly supports this claim is the increase in frequency of oblique cell walls in the quintuple mutant in Figure 1. The localization of Myo8A-GFP at the cortical division site only indirectly supports the guidance model. The authors' interpretation that these cortical Myo8A motors can exert a pulling force on the actin filaments emanating from the phragmoplast is plausible, but not supported by any data. It is also important to emphasize that this mechanism is very different from that found in budding yeast where the myosin uses actin filaments of opposite orientation to pull the nucleus toward the bud neck.

The text should be modified to explain these differences better. It may also be helpful to point out that the authors postulate two different localizations of Myo8A: one at MT plus ends and one on the cortical division site. Where it is stated that “...these data suggest that mysosin VIII steers the phragmoplast to the pre-determined cortical division site.”; at this point in the paper, it may be better to leave the window for mechanistic possibilities a little open since it might also be possible that myosin VIII could help to position other division factors, which would be a somewhat different role than being part of the steering mechanism itself.

1a) A major emphasis in the proposed mechanism for phragmoplast guidance involves the role of peripheral MTs. Several reviewers were not convinced by the data regarding the rate of incorporation of peripheral microtubules into the phragmoplast body; data currently consist of several movies, but these lack measurements, numbers of samples and statistics. Given that these data speak to the proposed mechanism, and the images already seem to have been acquired, quantitative comparisons of mutant and wild type (at comparable stages of cytokinesis) should be made.

1b) Towards the end of the Results section, is it known at this stage that myosin motility per se is the responsible activity? Other motors can act, for example, by localizing other proteins to particular positions such as the ends of cytoskeletal polymers.

2) Improve quantitation and statistical analysis of data: specifically

2a) While the histograms of cell wall angles in Figure 1 show different distributions among the three genotypes, it would add rigor to perform a statistical test for distribution shifts. For example, a Mann-Whitney U test.

2b) Please provide numbers for how often the population of cortical Myo8A-GFp spots are observed in the apical cells? They are evident in Figure 5, but not in 5D at the same time point.

2c) The FM-464 results are presented as single images. Numbers should be provided to indicate repeatability.

2d) It is stated that in the absence of actin, the peripheral microtubules with Myo8A on their ends are not rapidly incorporated into the expanding phragmoplast. The supporting movie does seem to show more stable association of microtubule ends with the cortex, and it is stated that they stay so associated for more than a minute; however, measurements really have to be provided to support this conclusion.

2e) The observations about actin becoming more abundant at the remaining unattached area of the phragmoplast need to be quantified – at least in terms of how often this was observed out of how many asymmetric cortical contact events.

2f) The experiment in Figure 5 should either be repeated on the line expressing LA-mCherry to demonstrate the extent of actin depolymerization by LatB, or other information presented to indicate the confidence that this treatment was effective as LatB can quickly lose activity during storage.

3) Improve display of data, including using kymographs, specifically:

3a) Taken altogether, Figure 2 and the accompanying movies are a little limited as data demonstrating Myo8A moving along actin tracks, at least as presented. There is that one nice example in Figure C, but some other examples should also be provided. A kymograph approach might work, where analysis lines are drawn along actin bundles and then applied to the Myo8A channel. This could work for Movie 1/image sequences of LF and Myo8A as it is possible then to show many motility events and to measure velocities. Performing a frame average across the time series in both the control and the LatB treatment in Movie 2 could make images much more distinctly different than as depicted in 2D.

3b) In Figure 3, and also Figure 5, is it possible to show projections along the plane of, or at an angle to, the phragmoplast equator? Or provide 3D rotations as movies? In both cases, one of the key features being pointed out is the population of tagged Myo8A at the periphery of the phragmoplast. While this location is clear in images such as 5C and E at 16 minutes and later, when the phragmoplast is tilted relative to the axis of view, it is less easy to discern at earlier stages (how much is peripheral and how much is internal to the periphery). In Figure 5 12 minutes, it is not in fact clear if the most intense portion of Myo8A ring is at the phragmoplast periphery, or somewhat external to this location.

3c) In Figure 5, show the two channels. It is important when assessing a colocalization pattern that both the merged and the single channel data are shown.

3d) The Myo8A-GFP localization to the presumptive plus ends of phragmoplast microtubules is interesting and the movies show the dynamic colocalization of these labels. To present such evidence in print, a kymograph analysis of joint displacement would be more effective.

3e) Movie 8 is a very important movie. Both individual channels should be shown together with the merged channels so dynamic co-localization can be better assessed.

3f) Peripheral microtubules are observed to be incorporated into the expanding phragmoplast within 10 seconds of what event?

4) Justify use of tobacco to examine the role of myo8s in plants with PPBs.

Testing models in a different plant is an important addition, but since P.p. myo8 was used instead of endogenous tobacco myo8, these results should be treated with caution. Showing that a myosin VIII from moss can show similar localization patterns when expressed in tobacco is not equivalent to showing the localization of a bonafide tobacco protein, let alone testing for the function. Two modifications should be made: First, an alignment of the P.p Myo8 sequence compared with tobacco (or other solanaceous species) should be included as a supplemental figure and second, changing the phrasing of the last sentence of the paragraph describing the tobacco results to be less emphatic. Perhaps “...myosin 8A is capable of...”

5) Explore phenotypes of mutants (or justify why mutants were not used) of Myo8 and formin in phragmoplast guidance.

Myo8-GFP only partially rescues the mutant. Why doesn't it rescue? One hypothesis is that it is just one of several myo8s and so we wouldn't expect it to completely rescue. The other is that the GFP tag interferes with its activity. Some evidence that it is not the latter, for example, by rescuing other myo8 lines (triple mutants?) with milder defects, would be useful. At the same time, since the Myo8-GFP doesn't fully rescue, this is an opportunity to see localization in a “mutant”. Is there any difference in behavior in the phragmoplasts that are not correctly aligned than in those that are? Formin is invoked as an important component of the division process; could mutants have been analyzed to strengthen the connection?

6) Clarify final model.

In the current model drawing, Formin is missing, which seems an important component, and the diagram implies delivery of Myo8A to the cortex by peripheral phragmoplast microtubules. Is there any evidence for this? It was reported that Myo8A-gFP shows up at the cortex before prophase, when there are no such microtubules.

---

## [Author Response]

*1) While the models presented in this manuscript are thought provoking, they are still quite speculative, and the text and title should be modified to reflect the fact that there is limited evidence for the guidance of the phragmoplast to the cortical division site*.

We have modified the title and text accordingly.

*The only result that directly supports this claim is the increase in frequency of oblique cell walls in the quintuple mutant in*
Figure 1*. The localization of Myo8A-GFP at the cortical division site only indirectly supports the guidance model. The authors' interpretation that these cortical Myo8A motors can exert a pulling force on the actin filaments emanating from the phragmoplast is plausible, but not supported by any data. It is also important to emphasize that this mechanism is very different from that found in budding yeast where the myosin uses actin filaments of opposite orientation to pull the nucleus toward the bud neck*.

*The text should be modified to explain these differences better. It may also be helpful to point out that the authors postulate two different localizations of Myo8A: one at MT plus ends and one on the cortical division site*.

We agree that at this point we have not provided direct evidence for our model. We hope that with the revisions to the text and model, the model is now presented more as a discussion point rather than a validated mechanism for guidance of plant cell division. In the revised model, we include the localization of formin to the phragmoplast midzone. Formins are known to generate actin filaments and stay associated with the barbed ends of the actin filaments. Thus, in our proposed model myosin VIII guides the peripheral phragmoplast microtubules along actin filaments towards the phragmoplast edge, the location of the actin filament barbed ends. With these revisions, we hope to clarify that our model is in agreement with the budding yeast mechanism whereby the plus end of the cytoplasmic microtubule is guided by myosin V, which walks to the barbed end of actin filaments located at the bud tip.

*Where it is stated that “...these data suggest that mysosin VIII steers the phragmoplast to the pre-determined cortical division site.”; at this point in the paper, it may be better to leave the window for mechanistic possibilities a little open since it might also be possible that myosin VIII could help to position other division factors, which would be a somewhat different role than being part of the steering mechanism itself*.

Thank you for pointing this out. We have modified the text accordingly.

*1a) A major emphasis in the proposed mechanism for phragmoplast guidance involves the role of peripheral MTs. Several reviewers were not convinced by the data regarding the rate of incorporation of peripheral microtubules into the phragmoplast body; data currently consist of several movies, but these lack measurements, numbers of samples and statistics. Given that these data speak to the proposed mechanism, and the images already seem to have been acquired, quantitative comparisons of mutant and wild type (at comparable stages of cytokinesis) should be made*.

We have performed the quantification as suggested and these new data are incorporated into the revised text.

*1b) Towards the end of the Results section, is it known at this stage that myosin motility per se is the responsible activity? Other motors can act, for example, by localizing other proteins to particular positions such as the ends of cytoskeletal polymers*.

Thank you. We have modified the text accordingly.

2) Improve quantitation and statistical analysis of data: specifically

*2a) While the histograms of cell wall angles in*
Figure 1
*show different distributions among the three genotypes, it would add rigor to perform a statistical test for distribution shifts. For example, a Mann-Whitney U test*.

We performed the Mann-Whitney U test and report the statistics in the revised figure legend.

*2b) Please provide numbers for how often the population of cortical Myo8A-GFp spots are observed in the apical cells? They are evident in*
Figure 5*, but not in 5D at the same time point*.

In apical cells, we always observe Myo8A-GFP accumulation at the cell cortex at anaphase. However the accumulation is transient and is not always captured in the still images from a particular time series.

*2c) The FM-464 results are presented as single images. Numbers should be provided to indicate repeatability*.

We have modified this figure (now Figure 7). The new figure shows images from a time series and the accompanying movies demonstrate that FM4-64 deposition is altered in the myosin VIII null and wild type treated with LatB. In addition, we included how often we observed uniform/non -uniform FM4-64 staining in wild type, Δmyo8ABCDE, and wild type treated with LatB.

*2d) It is stated that in the absence of actin, the peripheral microtubules with Myo8A on their ends are not rapidly incorporated into the expanding phragmoplast. The supporting movie does seem to show more stable association of microtubule ends with the cortex, and it is stated that they stay so associated for more than a minute; however, measurements really have to be provided to support this conclusion*.

Please see response to 1a.

*2e) The observations about actin becoming more abundant at the remaining unattached area of the phragmoplast need to be quantified* – *at least in terms of how often this was observed out of how many asymmetric cortical contact events*.

We removed these statements because we have new data with higher time resolution showing microtubules intersecting actin filaments between the leading edge of the phragmoplast and the cell cortex before incorporating into the expanding phragmoplast. These data, which were collected at a much higher frame rate (3.75 frames/sec, compared to 0.1 frames/sec), are strong evidence for microtubule translocation along actin filaments towards the phragmoplast leading edge. Accompanying supplementary movies are also provided and the best way to visualize these new data. The revised text describes these data.

*2f) The experiment in*
Figure 5
*should either be repeated on the line expressing LA-mCherry to demonstrate the extent of actin depolymerization by LatB, or other information presented to indicate the confidence that this treatment was effective as LatB can quickly lose activity during storage*.

Thank you. We have provided new data in Figure 5—figure supplement 1.

3) Improve display of data, including using kymographs, specifically:

*3a) Taken altogether,*
Figure 2
*and the accompanying movies are a little limited as data demonstrating Myo8A moving along actin tracks, at least as presented. There is that one nice example in Figure C, but some other examples should also be provided. A kymograph approach might work, where analysis lines are drawn along actin bundles and then applied to the Myo8A channel. This could work for Movie 1/image sequences of LF and Myo8A as it is possible then to show many motility events and to measure velocities. Performing a frame average across the time series in both the control and the LatB treatment in Movie 2 could make images much more distinctly different than as depicted in 2D*.

Thank you for the suggestions. We quantified the velocity of myosin VIII in moss cells and show the data in the revised Figure 2 and report the number in the revised text. Additionally, we show sample kymographs in supplement 1 to Figure 2. We have also provided frame averages, as suggested by the reviewers in the new Figure 2.

*3b) In*
Figure 3*, and also*
Figure 5*, is it possible to show projections along the plane of, or at an angle to, the phragmoplast equator? Or provide 3D rotations as movies? In both cases, one of the key features being pointed out is the population of tagged Myo8A at the periphery of the phragmoplast. While this location is clear in images such as 5C and E at 16 minutes and later, when the phragmoplast is tilted relative to the axis of view, it is less easy to discern at earlier stages (how much is peripheral and how much is internal to the periphery). In*
Figure 5
*12 minutes, it is not in fact clear if the most intense portion of Myo8A ring is at the phragmoplast periphery, or somewhat external to this location*.

We have provided still images from 3D rotations in a supplement to Figure 3 and Figure 5.

*3c) In*
Figure 5*, show the two channels. It is important when assessing a colocalization pattern that both the merged and the single channel data are shown*.

We have now split Figure 5 into two figures (Figure 5 and Figure 6). This allowed us to show separate channels for all the data in this figure. Thank you for the suggestion.

*3d) The Myo8A-GFP localization to the presumptive plus ends of phragmoplast microtubules is interesting and the movies show the dynamic colocalization of these labels. To present such evidence in print, a kymograph analysis of joint displacement would be more effective*.

While we have tried to do this, it has been very difficult to get a clear kymograph. This may be due in part to the fact that the peripheral phragmoplast microtubules are very faint in comparison to the rest of the phragmoplast. Such a faint signal is difficult to boost and does not appear well in the kymographs. We hope that the movies that are provided with the paper will be sufficient to show this.

*3e) Movie 8 is a very important movie. Both individual channels should be shown together with the merged channels so dynamic co-localization can be better assessed*.

Thank you for the suggestion. We have now incorporated the individual channels as well as the merge.

3f) Peripheral microtubules are observed to be incorporated into the expanding phragmoplast within 10 seconds of what event?

Based on the quantification that was requested (see 1a above), this line was removed from the text.

*4) Justify use of tobacco to examine the role of myo8s in plants with PPBs*.

Testing models in a different plant is an important addition, but since P.p. myo8 was used instead of endogenous tobacco myo8, these results should be treated with caution. Showing that a myosin VIII from moss can show similar localization patterns when expressed in tobacco is not equivalent to showing the localization of a bonafide tobacco protein, let alone testing for the function. Two modifications should be made: First, an alignment of the P.p Myo8 sequence compared with tobacco (or other solanaceous species) should be included as a supplemental figure and second, changing the phrasing of the last sentence of the paragraph describing the tobacco results to be less emphatic. Perhaps “...myosin 8A is capable of...”

As a supplement to Figure 4, we now provide a full alignment of the myosin VIII sequences from moss, *Nicotiana benthamiana*, and *Arabidopsis thaliana*. We also provide a table showing the percent identity among these myosin VIII proteins. We have modified the text and have also reworded the conclusions as suggested by the reviewers.

*5) Explore phenotypes of mutants (or justify why mutants were not used) of Myo8 and formin in phragmoplast guidance*.

Myo8-GFP only partially rescues the mutant. Why doesn't it rescue? One hypothesis is that it is just one of several myo8s and so we wouldn't expect it to completely rescue. The other is that the GFP tag interferes with its activity. Some evidence that it is not the latter, for example, by rescuing other myo8 lines (triple mutants?) with milder defects, would be useful. At the same time, since the Myo8-GFP doesn't fully rescue, this is an opportunity to see localization in a “mutant”. Is there any difference in behavior in the phragmoplasts that are not correctly aligned than in those that are? Formin is invoked as an important component of the division process; could mutants have been analyzed to strengthen the connection?

We agree with the reviewers that mutants in myosin would be a fantastic way to test our model. Unfortunately, we have not yet been able to characterize the biochemical properties of Myo8A. Without this, any mutations that we might make are potentially difficult to interpret, since we will not know how these mutations affect the properties of the myosin. While myosins are highly conserved, not all mutations have the same effect on all myosins. We would need substantial biochemical data to help support any mutational data.

We actually don’t expect full rescue of the Δmyo8ABCDE mutant with just Myo8A. From our previous work (Wu et al., Molecular Plant, 2011), we have shown that there are two groups of myosin VIIIs in moss. Myo8A is the myosin that is predominately expressed from its group, but Myo8B and 8C are in the other group. We expect that full rescue will likely be achieved when either Myo8B or Myo8C is also expressed.

Rescue of the triple mutant is an excellent idea. Unfortunately we have not generated a triple mutant that is lacking myo8A (only have Δmyo8BCD). We also do not currently have fluorescent tubulin in the other myosin mutants. With respect to formin, we primarily used formin localization as a proxy for where actin polymerization was being initiated on the phragmoplast. While formins may be integral to the process, it is difficult to assess this, since there are nine formins in moss, two of which are absolutely essential for polarized growth.

We thank the reviewers for suggesting these great experiments and we hope to do them in the future, but feel that they are beyond the scope of this study.

*6) Clarify final model*.

*In the current model drawing, Formin is missing, which seems an important component, and the diagram implies delivery of Myo8A to the cortex by peripheral phragmoplast microtubules. Is there any evidence for this? It was reported that Myo8A-gFP shows up at the cortex before prophase, when there are no such microtubules*.

We have re-drawn, clarified the model, and included formins. We did not mean to imply that delivery of Myo8A-GFP to the cell cortex occurs via peripheral microtubules. The new model has removed that component. We have also re- written the Discussion section to discuss the model with respect to division only in apical cells, where accumulation of cortical Myo8A-GFP happens at anaphase.